# Gridded maps of geological methane emissions and their isotopic signature

Giuseppe Etiope [1,2], Giancarlo Ciotoli [3,1], Stefan Schwietzke [4], Martin Schoell [5]

[1] Istituto Nazionale di Geofisica e Vulcanologia, Roma Italy
[2] Faculty of Environmental Science and Engineering, Babes Bolyai University, Cluj-Napoca, Romania
[3] Istituto di Geologia Ambientale e Geoingegneria, CNR-IGAG, Roma, Italy
[4] Cooperative Institute for Research in Environmental Sciences, University of Colorado, Boulder, Colorado, USA, and NOAA Earth System Research Laboratory, Global Monitoring Division, Boulder, Colorado, USA.
[5] Gas-Consult Int., Pleasanton, California, USA

Correspondence to: Giuseppe Etiope (giuseppe.etiope@ingv.it)

**Abstract**

Methane ($CH_4$) is a powerful greenhouse gas, whose natural and anthropogenic emissions contribute ~20% to global radiative forcing. Its atmospheric budget (sources and sinks), however, has large uncertainties. Inverse modelling, using atmospheric $CH_4$ trends, spatial gradients and isotopic source signatures, has recently improved the major source estimates and their spatial-temporal variation. Nevertheless, isotopic data lack $CH_4$ source representativeness for many sources, and their isotopic signatures are affected by incomplete knowledge of the spatial distribution of some sources, especially those related to fossil (radiocarbon-free) and microbial gas. This gap is particularly wide for geological $CH_4$ seepage, i.e., the natural degassing of hydrocarbons from the Earth's crust. While geological seepage is widely considered a major source of atmospheric $CH_4$, it has been largely neglected in 3D inverse $CH_4$ budget studies given the lack of detailed a priori gridded emission maps. Here, we report for the first time global gridded maps of geological $CH_4$ sources, including emission and isotopic data. The 1°x1° maps include the four main categories of natural geo-$CH_4$ emission: (a) onshore hydrocarbon macro-seeps, including mud volcanoes, (b) submarine (offshore) seepage, (c) diffuse microseepage and (d) geothermal manifestations. An inventory of point sources and area sources was developed for each category, defining areal distribution (activity), $CH_4$ fluxes (emission factors) and its stable C isotope composition ($\delta^{13}$C-$CH_4$). These parameters were determined considering geological factors that control methane origin and seepage (e.g., petroleum fields, sedimentary basins, high heat flow regions, faults, seismicity). The global geo-source map reveals that the regions with the highest $CH_4$ emissions are all located in the northern hemisphere, in North America, the Caspian region, Europe, and in the East Siberian Arctic Shelf. The globally gridded $CH_4$ emission estimate (37 Tg yr$^{-1}$ exclusively based on data and modelling specifically targeted for gridding, and 43-50 Tg yr$^{-1}$ when extrapolated to also account for onshore and submarine seeps with no location specific measurements available) is compatible with published ranges derived by top-down and bottom-up procedures. Improved activity and emission factor data allowed to refine previously published mud volcanoes and microseepage emission estimates. The emission-weighted global mean $\delta^{13}$C-$CH_4$ source signature of all geo-$CH_4$ source categories is -48.5‰ to -49.4‰. These values are significantly lower than those attributed so far in inverse studies to fossil fuel sources (-44‰) and geological seepage (-38‰). It is expected that using these updated more $^{13}$C-depleted, isotopic signatures in atmospheric modelling will increase the top-down estimate of the geological $CH_4$ source. The geo-$CH_4$ emission grid maps can now be used to improve atmospheric $CH_4$ modelling, thereby improving the accuracy of the fossil fuel and microbial components. Grid csv files are available at https://doi.org/10.25925/4j3f-he27.

## 1. Introduction

Methane ($CH_4$) is a powerful greenhouse gas, whose concentrations in the atmosphere increased about 2.5 times since the pre-industrial era (1750), approaching 1.9 ppm in 2018. With a global emission of about 558 Tg $CH_4$ $yr^{-1}$ (Saunois et al., 2016), $CH_4$ contributes ~20% to global radiative forcing (Ciais et al. 2013). The $CH_4$ budget, i.e. natural and anthropogenic sources and sinks, estimated by either bottom-up (emission inventories and process-based models) or top-down (inverse modelling) approaches (e.g., Saunois et al., 2016 and Refs. therein), is subject to considerable uncertainties, however. With respect to natural sources, top-down estimates show strong disagreement with bottom-up estimates, both globally and regionally. Global box-modelling based on isotopic measurements (stable C isotope ratio, $\delta^{13}$C-$CH_4$) of source signatures and the atmosphere combined with three-dimensional (3D) forward modelling using trends and spatial gradients recently improved the knowledge of major sources (fossil-fuel, agriculture and wetlands) and their spatio-temporal variation (e.g., Schwietzke et al 2016). Nevertheless, isotopic data lack representativeness of $CH_4$ source signatures for many sources, and source attributions are limited by incomplete knowledge of the spatial distribution of some major sources, especially fossil fuel and microbial. In this respect, geological $CH_4$ emissions, i.e. the natural component of fossil fuel ($^{14}$C-free) emission, play a critical role. Geological $CH_4$ sources are the natural degassing of hydrocarbons from the Earth's crust (e.g., Etiope and Klusman, 2002; Kvenvolden and Rogers, 2005; Etiope, 2015). Geo-$CH_4$ originated in deep rocks by biotic (i.e. microbial and thermogenic) processes related to petroleum fields in sedimentary basins, as described in a wide petroleum geology literature (see for example Etiope, 2017 for a recent overview). Relatively minor amounts of $CH_4$ can also be produced by abiotic processes, which do not involve organic matter in rocks (e.g., magma degassing, high temperature post-magmatic process, $CO_2$ hydrogenation or Sabatier reaction, in geothermal/volcanic systems and ultramafic igneous rocks; e.g., Etiope and Sherwood Lollar, 2013). Surface emissions of geological $CH_4$ occur through the process known as "gas seepage", which includes point sources (gas-oil seeps, mud volcanoes, springs, geothermal manifestations) and area sources (diffuse "microseepage"). Once considered a minor natural $CH_4$ source globally (e.g., Lelieveld et al., 1998; Prather et al., 2001), geological degassing is today recognised as a major contributor to atmospheric $CH_4$, as indicated by the agreement between bottom-up and top-down estimates converging to 40-60 Tg $yr^{-1}$ (Etiope et al. 2008; Ciais et al. 2013; Etiope, 2015; Saunois et al. 2016; Schwietzke et al. 2016). Nevertheless, geological seepage has mostly been neglected in global top-down $CH_4$ budget studies (e.g., Bousquet et al. 2006; Bergamaschi et al. 2013). In addition, geological $CH_4$ has erroneously been considered to be typically $^{13}$C-enriched, thus with relatively high $\delta^{13}$C-$CH_4$ values compared to biological sources such as wetlands (a global average of -38‰ was assumed for seepage by Sapart et al. 2012). In petroleum geochemistry it is well known, in fact, that in addition to the common thermogenic gas produced by moderate to high maturity source rocks, typically with $\delta^{13}$C-$CH_4$ from -30‰ to about -50‰, vast amounts of methane in sedimentary basins are microbial (thus with $\delta^{13}$C-$CH_4$ ranging from -55 to about -90‰) and thermogenic from low maturity source rocks, with $\delta^{13}$C-$CH_4$ from -50‰ to about -70‰ (Etiope, 2017; Milkov and Etiope, 2018). Degassing (seepage) to the atmosphere of $^{13}$C-depleted geo-$CH_4$ sources is also widely documented (e.g., Etiope et al. 2009 and references therein). In addition to using unrepresentatively heavy

$\delta^{13}$C-CH$_4$ geo-CH$_4$ values in previous studies, detailed a priori gridded maps of geo-CH$_4$ emissions and its isotopic signatures, which are essential for 3D inverse modelling and to discriminate between natural and anthropogenic microbial emissions, are currently lacking.

Here, we report the first global grid maps of geological CH$_4$ sources, including emissions and isotopic source signatures. The maps, elaborated by ArcGIS at 1°x1° resolution, include the four main categories of natural geological CH$_4$ sources: (a) onshore hydrocarbon macro-seeps (including mud volcanoes), (b) submarine (offshore) seeps, (c) diffuse microseepage and (d) geothermal manifestations. For each category we have developed an inventory of point sources and area sources, including coordinates (areal distribution, i.e. activity), estimated CH$_4$ fluxes (emission factors) and $\delta^{13}$C-CH$_4$ values. These parameters have been determined considering several geological factors that control CH$_4$ origin and seepage (petroleum fields, sedimentary basins, faults, earthquakes, geothermal/volcanic systems), based on published and originally *ad-hoc* developed datasets, as described in Sections 4, 5, 6 and 7. Integrated (total geo-CH$_4$) maps and associated text files (csv, comma- separated-values) have been generated to facilitate atmospheric CH$_4$ modelling to improve the accuracy of fossil fuel and microbial components. Gridded geo-CH$_4$ emissions were compared with published global estimates, derived by different approaches (e.g., Etiope et al. 2008; Etiope, 2012; 2015; Schwietzke et al. 2016). Gridded emissions do not necessarily represent the actual global geo-CH$_4$ emission or improve previous estimates, because the datasets developed for the gridding may not be complete or may not contain all the information necessary for improving previous estimates. A refinement of bottom-up estimates has only been possible for mud volcanoes and microseepage, because their gridding implied a careful assessment of the spatial distribution and emission factors.

## 2. Classification of the geological CH$_4$ sources

Geological CH$_4$ sources can be classified into four major categories:

(a) Onshore hydrocarbon seeps (or macro-seeps) in sedimentary (petroliferous) basins including CH$_4$-rich gas-oil seeps, mud volcanoes (MV) and gas-bearing springs. Hereafter referred as **OS**.

(b) Submarine (offshore) seeps, where CH$_4$ released from shallow seafloor (coastal areas or shelves, generally up to 300-400 m below sea level) can cross the water column and enter the atmosphere. Hereafter referred as **SS.** Deep-sea seeps that are unlikely responsible for methane emission into the atmosphere are not considered.

(c) Diffuse microseepage in sedimentary (petroliferous) basins, the widespread, invisible exhalation of CH$_4$ typically detected in correspondence with gas-oil fields. Hereafter referred as **MS.**

(d) Geothermal and volcanic manifestations, where CH$_4$ is a minor component (subordinated to CO$_2$) but with potentially significant fluxes to the atmosphere. Hereafter referred as **GM.**

These "geo-methane" sources are extensively described and discussed in a wide body of literature; for details and definitions the reader may refer to Etiope and Klusman (2002); Judd (2004); Kvenvolden and Rogers (2005); McGinnis et al (2006); Etiope et al. (2007); Judd and Hovland (2007); Etiope et al. (2008); Etiope and Klusman (2010); Etiope (2015), Mazzini and Etiope (2017). Their global bottom-up and top-down emissions, compared with other natural CH$_4$ sources, are summarized in Fig.1.

**3. Methodology**

Methods for creating $CH_4$ emission and $\delta^{13}C$-$CH_4$ grids vary by geo-$CH_4$ category, based on the data

availability and specific seepage characteristics. Methods are therefore described in detail for each category

in Sections 4 (OS), 5 (SS), 6 (MS) and 7 (GM). First, a brief overview of the different types of input data and

gridding procedure is given below.

*3.1 Data sources*

Table 1 summarizes how the four categories of geo-$CH_4$ sources were elaborated, showing data sources,

the parameters used to define the "activity" (spatial distribution), the "emission factors" (fluxes), and the

attribution of the isotopic $CH_4$ values. The list and web links of the sources of databases are reported in the

Supplement (S6).

*3.2. Gridding procedure*

The gridding procedure is the same for each geo-$CH_4$ source category. Geo-$CH_4$ emission and isotope

datasets were imported in ArcGIS environment and saved in either point (OS and GM) or polygon (SS and

MS) shapefile format, including coordinates and attributes (i.e., type of emission, area, emission factor,

isotopic $CH_4$ values, plus geographical information, such as country and region). The grid was then joined

with single OS, SS, MS, GM shapefiles. The final csv files include data fields that define the coordinates of

each cell centroid, the variable name and its unit of measurement (tonnes year$^{-1}$ per cell for $CH_4$ emission

and ‰ for $\delta^{13}C$-$CH_4$, according to VPDB, Vienna Pee Dee Belemnite, standard). For the grid cell values, the

number zero (0) is used to indicate an actual or best emission estimate of zero (no seepage), whereas -

9999 indicates lack of knowledge, of the emission. Specifically:

In the $CH_4$ output files:

- zero (0) value is used for:

- all offshore cells of the onshore seepage shape files (OS, MS and GM)

- all onshore cells of the offshore seepage (SS).

- all onshore cells outside the potential MS area

- onshore cells without OS or GM sources

- offshore cells outside the SS areas

- the number -9999 is used for:

- cells within SS areas where emissions are unknown.

The categories OS, MS and GM, due to the method of emission derivation (see related sections below) have

always an emission value.

In the isotope files:

- an isotopic value is reported in each cell that has a flux value;

- where specific values are not available (as occurred in OS and SS), the global weighted average $\delta^{13}C$ for the relative emission category is reported;
- four decimals are used for global weighted average isotope values; this can help to trace back which cells are based on cell-specific data (with one decimal), and which contain weighted averages (four decimals);
- the value -9999 is used only for cells with no emissions in the corresponding $CH_4$ output files.

The application of such rules is described in the specific chapters of the four emission categories. Once individual OS, SS, MS, GM maps/files were produced, they were merged into a unified, total geo-$CH_4$ gridding: emissions per cell were summed and $\delta^{13}C$ values were averaged.

## 4. Onshore seeps (OS)

### 4.1 Global seep count and distribution

The spatial distribution (activity) of onshore seeps is derived from geographic coordinates of 2827 seeps, from 89 countries, reported in a global onshore seep dataset, which includes 1119 oil seeps, 846 gas seeps, 741 mud volcanoes and 121 gas-bearing springs. This dataset is an updated version of a previous inventory (named GLOGOS, reporting 2100 seeps) available from CGG (2015) and described in Etiope (2015). The global distribution of OS is reported in Fig. 2.

The seeps listed in the dataset generally refer to individual focused vents (single macro-seep manifestations) but in several cases they refer to groups or clusters, or even wide zones of multiple seep points. 612 seeps (569 gas-oil seeps and 43 mud volcanoes) could not be geographically located with precision and they are listed without coordinates (in addition to the 2827 seeps). The dataset, therefore, actually mentions a total of 3439 seeps or seepage sites, including 3396 gas-oil seeps and 784 mud volcanoes. The total number of 3439 OS represents about 30% of total seeps assumed to exist on Earth (≈10,000 was proposed by Clarke and Cleverly, 1991), but the present dataset includes the largest and more active seeps (especially for MV) because they are more easily documented and have attracted attention for scientific research, petroleum exploration, and natural heritage protection. Small or inactive seeps tend to be less observed and reported. In particular, the MV inventory is almost complete, probably missing smaller MVs in Asia. The gas-oil seeps in the dataset likely contribute more than 50% of the previously estimated total gas-oil seep emission. Africa and South America likely host a larger number of gas-oil seeps and springs not documented in the dataset, because of the paucity of specific investigations, especially in remote areas.

### 4.2 Attribution of $CH_4$ emissions to individual seeps

The attribution of $CH_4$ emission magnitudes to individual seep locations follows two different approaches for (a) gas-oil seeps or springs and (b) mud volcanoes (MV).

*4.2.1 Emission of gas-oil seeps and springs*

Direct measurements of $CH_4$ flux are available for about 100-200 gas-oil seeps in Europe, Asia and North America (see Table 6.1 in Etiope, 2015). In general, therefore, potential or theoretical flux values have been attributed to the inventoried seeps. Theoretical emission values can be reasonably provided only in terms of order of magnitude (i.e. $10^0$, $10^1$, $10^2$, $10^3$, $10^4$ tonnes year$^{-1}$). For gridding purposes, however, theoretical values (approximate working values) were used taking into account basic characteristics of the several seeps, i.e. the type of seep (for example, gas seeps generally release more methane than oil seeps), the activity and size of the seep (according to specific literature, reports, web images), and taking into account, as "calibration", experimental data, i.e., flux values measured in the field from seeps covering a wide range of activity and size (data are taken from the wide literature considered in Table 6.1 of Etiope (2015)). The theoretical values also take into account the gas emission from the invisible miniseepage, the diffuse degassing from the ground surrounding the macro-seep craters and vents (see definitions in Etiope, 2015), and which adds an amount of gas that may be three times higher than that released from the macro-seep (Etiope, 2015). This resulted in the attribution of the values reported in Table S1 in the Supplement. These values should be considered as first-order estimates and care should be taken when using individual seep flux estimates from this product to derive global emission estimates, as discussed in section 4.5.

*4.2.2 Emission of mud volcanoes (MV)*

For MV, emission values refer to the continuous quiescent degassing, i.e. they do not include emissions during episodic eruptions, as these are practically impossible to estimate for each MV. Eruptions were considered separately for the global emission estimate as discussed below. The quiescent emissions were attributed to each MV following the activity (area) and emission factor approach as follows.

A precise evaluation of the MV areas was performed by accurate image (Google Earth) analysis. For each MV visible on Google Earth images, the area of the entire MV structure, including central craters and flanks, was estimated by drawing a polygon encompassing the mud cover and mound flanks. For smaller MV, not visible on low resolution Google Earth images or covered by vegetation, photos or information from published literature or web sources were considered to define the order of magnitude of the MV size. From two repeated image analyses the global MV area resulted to be about 680±40 km$^2$.

The MV emissions were then assessed using an updated dataset of fluxes measured from 16 MV in Azerbaijan, Romania, Italy, Taiwan, China and Japan (Table S2 in Supplement), distinguishing between the macro-seepage (the focused emission from craters and vents) and miniseepage. Regression analysis between MV area, miniseepage and macro-seep flux of these measured MV was used to derive miniseepage and macro-seep flux (and thus the total $CH_4$ emission) for each MV of the inventory, whose area was determined as previously indicated. The procedure is described in detail in the Supplement (Section S1.1)

*4.2.3 The "big emitters"*

There is a total of 76 OS with emissions in the order of $10^4$ tonnes $CH_4$ yr$^{-1}$ (i.e. that may emit at least 10,000 tonnes $CH_4$ yr$^{-1}$ individually), and they can be considered "big emitters". They typically refer to large, very

active and frequently erupting MVs so their emission is estimated based on emission factor and area approach described in the previous section. The 76 big emitters likely dominate the spatial distribution of $CH_4$ emissions (they represent 63% of the total OS emission) and the weighted global mean isotopic value. As shown in Fig. 3, it is clear that, on a global scale, the Caspian and Mid-East regions represent the main OS emission areas.

### 4.3 Attribution of the $\delta^{13}C$-$CH_4$ value

For each seep the $\delta^{13}C$-$CH_4$ value is:

(a) measured, as indicated in the literature (available in the OS inventory; CGG, 2015), or (b) estimated on the basis of isotopic values using one of the following three procedures, in priority order:

- of similar seeps occurring in the same basin (when these data are available)

- of reservoir gas in the same petroleum field, from Sherwood et al. (2017) dataset or literature

- suggested by local petroleum geology (existence of microbial gas, thermogenic gas, oil), when the previous procedures cannot be applied.

The OS emission-weighted value (Section 4.5.2) was used for gridding where the isotopic value could not be assessed. The global distribution of three classes of $\delta^{13}C$-$CH_4$ value is shown in Fig. S4 in the Supplement.

### 4.4 OS gridding

The OS shapefile generated in ArcGIS was spatially joined to the 1°x1° vector square grid. OS occur in 616 cells, for a total emission of 3.9 Tg yr$^{-1}$ (Fig. 4). This is about 0.1 Tg yr$^{-1}$ higher than the actual sum of the seep emission in the inventory because of multiple counting of 57 seeps that occur exactly on the boundary of a cell.

### 4.5 Evaluation of global OS emission and $\delta^{13}C$-$CH_4$

#### 4.5.1 Re-assessing global OS emission

Because the OS inventory is not complete and the uncertainty of the theoretical flux values considered for individual oil-gas seeps is large (see Section 4.6), the OS flux grid is not meant to update or refine the previous global OS $CH_4$ emission estimate (Etiope et al. 2008). A comparison with the published bottom-up estimates can establish whether the OS inventory data used in the OS flux grids are plausible. However, the procedure developed to attribute $CH_4$ emissions to MVs can represent a refinement of the global MV emission estimate.

Published bottom-up estimates of $CH_4$ emission from onshore macro-seeps are reported in Table 2. Some estimates included, without a clear distinction, shallow submarine MV (e.g., Dimitrov, 2003; Milkov et al. 2003), which must be considered within the category SS in this work. Therefore, those estimates are indicated in the table as upper limit. Because the data of the OS inventory, as explained in Section 4.2, refer

only to quiescent degassing, the table distinguishes emissions that exclude MV eruptions (quiescent degassing) and those that include MV eruptions.

Concerning gas-oil seeps and springs, the use of the theoretical values, as described in Section 4.2, results in global $CH_4$ emission of about 1 Tg yr$^{-1}$ (Table 2). As indicated in Section 4.1, the OS dataset, although representing only 30% of all seeps existing on Earth, includes the largest and more active seeps, which may contribute at least 50% of the global emission; accordingly, the total gas-oil seep emission could be likely around 2 Tg yr$^{-1}$. Any further or more detailed extrapolation to a global seep emission estimate would be inappropriate.

The global MV emission from quiescent degassing, i.e. the sum of the MV emission values reported in the OS dataset, amounts to ~2.8 Tg yr$^{-1}$. The total $CH_4$ emissions from the 2827 OS seeps is, therefore, about 3.8 Tg yr$^{-1}$ (1 + 2.8 Tg yr$^{-1}$). The OS-MV dataset likely represents about 90% of total MVs assumed to exist on Earth (≈900; Dimitrov, 2002; Etiope and Milkov, 2004); extrapolating to the total MV number would result in a global MV emission of approximately 3 Tg yr$^{-1}$. This is within the range suggested by Etiope and Milkov (2004). Compared to previous emission estimates of Etiope and Milkov (2004) and Etiope et al (2011), the present MV estimate used a lower activity, i.e. lower global area, 680 km$^2$ instead of 2800 km$^2$ (which was suggested by data provided by Azerbaijan Geological Institute) but relatively higher emission factors. Concerning the MV eruptions, we can only use, again, the rough estimations indicated in Dimitrov (2003), Milkov et al. (2003) and Aliyev et al. (2012) (i.e., average gas flux during eruptions of MVs in Azerbaijan 2.5x10$^8$ m$^3$, the proportions of eruptive MVs: 27%, and the frequency of eruption: 1.35 eruptions/year), which translate into a total eruptive emission of 3.1 Tg yr$^{-1}$ (Milkov et al. 2003). Therefore, the global OS emission, including MV eruptions and assuming the theoretical values for the gas-oil seeps and springs, would be ~ 8.1 Tg yr$^{-1}$, which is within 10% of the lower range proposed by Etiope et al. (2008).

*4.5.2 The average emission-weighted $\delta^{13}$C-CH$_4$*

The total mean value of $\delta^{13}$C-CH$_4$ from all OS is -47.8‰, and that from the 76 big emitters is -46.7‰. The global OS emission-weighted mean value of $\delta^{13}$C-CH$_4$ is -46.6‰.

### *4.6 OS uncertainties*

*Spatial distribution uncertainty:* In the 1°x1° grid, the uncertainty of the geographic distribution of the OS is practically zero, as all identified seeps have geographic coordinates within an error <1°.

*Emission uncertainty:* The uncertainty of the modeled oil-gas seep emission (based on the method of value attribution described in Section 4.2.1) is maximum 90% (1 ±0.9 Tg yr$^{-1}$). The uncertainty of global MV emission (48%) was estimated by summing (a) the uncertainty of the estimated MV areas (6%, see Section 4.2.2) and (b) the uncertainty of the modelled MV emission factor (42%; see Supplementary S1.1). Because oil-gas seeps and MVs account for 26% and 74% of total OS emission, respectively, the overall gridded OS emission uncertainty is about 58% (3.8 ±2.2 Tg yr$^{-1}$).

*$\delta^{13}$C-CH$_4$ uncertainty:* The uncertainty of measured $\delta^{13}$C values (from literature) practically corresponds to laboratory analytical uncertainty (typically <0.1‰). The maximum uncertainty of the estimated $\delta^{13}$C values

(based on criteria described in Section 4.3) is approximately within 15‰, i.e. half of the range of $\delta^{13}$C values for typical microbial (-80 to -60‰) and thermogenic (-50 to -20‰) gas. The uncertainty of the emission-weighted mean (-46.6‰) is mainly induced by the 76 big emitters, for which the $\delta^{13}$C values are available or estimated with good approximation (<±5‰), leading to a mean value of -46.7‰. The difference between global emission-weighted and 76 big emitters average $\delta^{13}$C values is 0.1‰. The average order of magnitude of ±1‰ can be considered for the uncertainty of global emission-weighted $\delta^{13}$C value.

## 5. Submarine seepage (SS)

### 5.1 Assessment of global SS area

A specific dataset of offshore seepage areas, in coastal regions and shallow seas (typically <500 m deep, which is generally the maximum depth of seeps that may affect the atmosphere; e.g., Solomon et al. 2009), was developed based exclusively on published literature (Table S4 in the Supplement). The dataset includes:

a) Submarine seeps (including mud volcanoes) where gas was observed to reach the sea surface via bubble plumes and the emission to the atmosphere was estimated: flux emission estimates are available from 15 zones (from focused, point-source, manifestations to wide regional areas) in the seas of Australia, Bulgaria, Brunei,, Canada, Chile, China, Denmark, Georgia, Greece, Norway, Spain, Romania, Russia, Turkey, Ukraine, United Kingdom, USA.

b) Submarine seeps in shallow seas where gas was actually observed (also through hydro-acoustic images) to reach the surface but the output to the atmosphere was not provided, or where, due to the shallow seabed (<400-500 m), the methane is expected to enter the atmosphere. These areas (16 zones) are in the offshore of USA, Canada, Mexico, The Netherlands, Denmark, France, Italy, Greece, Russia, Azerbaijan, Turkmenistan and Pakistan.

The dataset does not include deep-sea seeps or areas with gas-charged sediments (e.g. as those inventoried by Fleischer et al. 2001) that may release methane into the water column, but for which the possibility of injection into the atmosphere is scarce or unknown. The area and methane flux estimates reported in the several papers were used here without critical evaluation. Geo-referenced polygons were created for each area (Fig. 5).

### 5.2 Attribution of seepage levels

$CH_4$ fluxes from the original publications (Table S4) are used in the gridded emission dataset.

### 5.3 Attribution of the $\delta^{13}$C value

The $\delta^{13}$C-$CH_4$ values of SS are attributed on the basis of available literature or considering the geological

setting (type of petroleum system, origin of the gas) of the seepage areas (*italic* values in Table S4)

following the same criteria adopted for OS. For four areas in Table S4 (China-Brunei offshore, Laurentian

Channel and Grand Banks Downing Basin in Canada, and East Kamtchatka shelf in Russia) it was not

possible to attribute any theoretical $\delta^{13}C$ value because the gas may actually derive from either microbial or

thermogenic sources, covering a wide range of isotopic values. In these cases, the global emission-

weighted $\delta^{13}C$ value of SS (see Section 5.5) was used for these regions in the $\delta^{13}C$ grids. The global map of

$\delta^{13}C$ for SS is shown in Fig. S5 in the Supplement.

### *5.4 SS gridding*

The SS grid dataset was generated digitizing polygons of the SS areas from literature maps (see references

in Table S4). The final shapefile contains 31 polygons characterized by the following variables: country,

longitude and latitude of the polygon centroid, $CH_4$ output flux, area, and average $\delta^{13}C$ value of the

emissions in each polygon. The value -9999 is reported for the missing emissions at 16 sites (sites 16 to 31

in Table S4). The SS layer was joined with the 1°x1° vector grid and the resulting map is shown in Fig. 6.

### *5.5 Evaluation of global SS emission and $\delta^{13}C$-CH$_4$*

The sum of $CH_4$ emissions from the 15 SS areas in Table S4 (which refer to published estimates) is ~3.9 Tg

$yr^{-1}$. This represents about 20% of the theoretical estimate of global SS emission to the atmosphere (~20 Tg

$yr^{-1}$), derived by process-based models, proposed by Kvenvolden et al (2001). SS emissions also occur in

the other 16 areas reported in Table S4 and likely in many other sites not investigated yet. Among the areas

with missing emission values, the Gulf of Mexico and the Caspian Sea are very likely major methane

emitters, followed by the North US Atlantic margin. It is difficult to evaluate whether adding these missing SS

emissions, the total sum would approach the Kvenvolden et al. estimate of 20 Tg $yr^{-1}$ (however, we consider

this global value as a theoretical reference for our SS gridded emission, not a target or actual value to

reach). Evaluation of the SS emission factor (based on the reported area and total fluxes in Table S4) is also

difficult because the areas indicated in the several works (see References in Table S4) often refer to the

surveyed area and not to the actual area of seepage; in these cases, using the surveyed area would result

in a strongly underestimated emission factor. However, using the relationship observed for the 15

"investigated" sites between area (actual seepage or surveyed) and emission factor, the other 16 sites would

yield total emissions of about 1 Tg $yr^{-1}$. This would bring the total $CH_4$ emission from the 15+16 sites of

Table S4 to about 5 Tg $yr^{-1}$. Assuming that (a) SS generally do not take into account the release of dissolved

methane (i.e., only methane bubbles are accounted for) and (b) today unknown SS areas (not listed in Table

S4) may have a seepage extent not exceeding that of the investigated areas, it is plausible that global SS

emission exceeds 5 Tg $yr^{-1}$. If the upper estimate for the East Siberian Arctic Shelf, 4 Tg $yr^{-1}$ (Berchet et al.,

2016; i.e., twice the mean used in Table S4), is considered, then the global SS emission would exceed 9 Tg

$yr^{-1}$. The SS emission-weighted mean value of $\delta^{13}C$-CH$_4$ is -59‰. The non-weighted mean value is -51.2‰.

### 5.6 SS uncertainties

*Spatial distribution uncertainty:* In the 1°x1° grid, the uncertainty of the geographic distribution of the SS is practically zero, as all seepage zones have geographic coordinates within an error <1°.

*Emission uncertainty*: The main uncertainty and control on the global gridded 3.9 Tg yr$^{-1}$ value is associated with the estimate of $CH_4$ emissions from the East Siberian Arctic Shelf, for which we used the central value (2 Tg yr$^{-1}$) of the range indicated by Berchet et al (2016), i.e. 0-4 Tg yr$^{-1}$ (not very dissimilar from the estimate of 2.9 Tg yr$^{-1}$ suggested by Thornton et al. 2016). The other 15 SS areas, totaling ~1 Tg yr$^{-1}$, have variable uncertainty, often not defined in the individual publications. With a ±2 Tg yr$^{-1}$ uncertainty for the Siberian Arctic Shelf and assuming arbitrarily 10% uncertainty for the other estimates, the overall SS gridded emission uncertainty would result ±2.1 Tg yr$^{-1}$ (54%).

*$\delta^{13}$C-$CH_4$ uncertainty:* The maximum uncertainty of the estimated $\delta^{13}$C values (based on criteria described in Section 5.3) is approximately within ±15‰, i.e. half of the range of $\delta^{13}$C values for typical microbial (-80 to -60‰) and thermogenic (-50 to -20‰) gas. The uncertainty of the emission-weighted mean (-59‰) is mainly controlled by emissions from Eastern Siberian Arctic Shelf, North Sea and Black Sea, whose $\delta^{13}$C values are available or estimated, ranging from -50 to -63‰. The overall uncertainty of the global emission-weighted mean is thus reasonably < ±7‰.

## 6 Microseepage (MS)

### 6.1 Assessment of global MS area

The diffuse exhalation of $CH_4$, called microseepage (MS), is widespread throughout onshore petroleum fields all over the world. It is systematically observed in correspondence with anticlines and marginal (faulted) areas of gas-oil fields (Etiope and Klusman, 2010; Tang et al. 2017). The existence of macro-seeps (OS) in a given region also implies a high probability that MS exists in that region, even if that region falls outside a known petroleum field. Therefore, as a proxy of the activity (spatial distribution) of MS, we considered the global area of petroleum fields and a global area including OS defined as described below. This criterion is conservative as MS may also occur in sedimentary basins without known petroleum fields and OS (Klusman et al. 2000; Etiope and Klusman, 2010). The assessment of the global petroleum field area (PFA) and global OS area (OSA) is discussed in the Supplement (Sections S3.1 and S3.2). The total potential MS area (PMA) resulted to be PFA + OSA = 13,033,000 + 85,900 = 13,118,900 km$^2$.

34

35

### 6.2 Attribution of MS levels

The level of MS $CH_4$ emissions was established on the basis of a statistical analysis of a MS flux data-set (see Section 6.2.1) and considering the theory of seepage migration mechanisms, for which the gas flux

greatly depends on the permeability of the rocks, especially when induced by faults and fracture networks (Etiope and Klusman, 2010; Etiope, 2015; Tang et al. 2017). Accordingly, the attribution of the flux within the PMA (PFA+OSA) was done considering the presence/absence, in each cell, of three major geological factors, which are proxies of methane seepage and gas permeability, i.e. OS, faults and seismicity, as explained in Section 6.2.2.

*6.2.1 Statistics of MS data*

A dataset of 1509 MS $CH_4$ flux measurements was compiled based on available literature and unpublished works. The data are from 19 petroleum areas: 8 in the USA (Klusman et al 2000; Klusman, 2003; Klusman, 2005; Klusman, unpublished; LTE, 2007), 6 in Italy (Etiope and Klusman, 2010; Sciarra et al. 2013; Etiope, 2005; Etiope, unpublished), 1 in Romania (Etiope, 2005), 1 in Greece (Etiope et al. 2006; Etiope, unpublished), and 3 in China (Tang et al. 2007; 2010; 2017). The resulting descriptive statistics are reported in Table S5 in the Supplement.

The data are divided into two groups: (a) negative and near-zero values ($<0.01$ mg m$^{-2}$ d$^{-1}$, considering minimum analytical error), which represent the normal $CH_4$ flux in dry (not flooded) soils, and (b) positive values, $>0.01$ mg m$^{-2}$ d$^{-1}$ (i.e. microseepage). The similar order of magnitude between the median and the geometric mean flux indicates a log-normal behavior of the positive $CH_4$ flux distribution. The positive values represent about 57% of total measurements. This implies that MS does not occur throughout the entire PFA. This is well known, as $CH_4$ flux from the ground, in addition to underground rock permeability and fluid pressures, depends also on soil conditions (humidity, porosity, temperature) and methanotrophic activity. Accordingly, and taking into account that the MS measured sites are geographically dispersed with a relatively homogeneous spatial distribution (and the measurements were taken in different seasons), we reduced the PFA by removing 43% of the area as described in Section 6.4. A new MS area was therefore defined as "Effective Microseepage Area", EMA, which is OSA + 57% of PFA. The derivation of the EMA area is described in Section 6.4. Frequency histogram and Normal Probability Plot (NPP) of MS data (logarithmic values of positive values) confirm that flux values have a log-normal distribution (Fig. S7). Values exceeding 1000 mg m$^{-2}$ d$^{-1}$ (up to 7078) were excluded as they represent special and rare cases of MS (often not distinguishable from miniseepage, which is the halo surrounding macro-seeps).

The combined analysis of NPP and frequency histogram (Fig. S7) resulted in the identification of 4 main groups of positive flux data, i.e. 4 levels of MS:

Level 1    0.01-12 (median: 1.3) mg m$^{-2}$ d$^{-1}$

Level 2    12-60 (median: 31.1)

Level 3    60-300 (median: 110)

Level 4    300-1000 (median: 493.5)

Level 0 implies absence of MS. The median of each level was assigned to the 0.05°x0.05° grid cells included in the area with expected MS defined in Section 6.2.2 and according to the presence of the factors influencing MS. The median was chosen because it is not affected by outliers within each level, providing conservative flux values.

*6.2.2 Factors influencing MS level: presence of macro-seeps, faults and seismicity.*

The 4 MS levels (1, 2, 3 and 4) are associated with 4 different combinations of the three factors influencing the gas flux, following MS theory and experimental data. The three factors are:

(a) faults;

(b) seismic activity;

(c) presence of macro-seeps (OS), which are themselves expression of regional seepage activity;

(a) Fault data were taken from 17 different datasets (see Sources of databases in the Supplement): the main one is the Global Faults layer of ArcAtlas (Finko, 2014). It includes two types of faults: (1) faults created by the dislocation of rocks that define the geological structures of the continents (tectonic contacts and thrust-faults) and (2) faults created by the morphology of the present-day relief and morphostructure (steps and rifts). The first type of faults refers to ancient structures, while those revealed by relief are comparatively young structures that appeared during the neotectonic stage of the Earth's evolution (mostly in the Neogene and Quaternary periods). The other 16 fault datasets are national or regional datasets from Afghanistan, Australia, Bangladesh, Caribbean region, Central Asia, Europe including Turkey, Georgia, Greece, Ireland, Italy, New Zealand, South America, Southern Mediterranean area, Spain, Switzerland, United Kingdom (see Supplement). The final merged fault dataset includes 156,095 tectonic elements (Fig. S8); obviously it does not include all actual existing faults on Earth. The dataset must be interpreted as a global distribution of the main regional fault systems and fractured zones.

(b) The epicenters of earthquakes are proxies of fault location and activity (permeability), so they also represent the presence of active faults, which may not be reported in the fault dataset. It is also known that gas migration and escape to the surface may increase with seismic activity. We used the seismicity dataset of USGS Earthquake Lists, Maps and Statistics (see Sources of databases in the Supplement). We extracted only onshore seismic events with magnitude M>4.5 recorded from 2005 to 2017. This resulted in a dataset of 18,157 onshore epicenters covering 177 countries (Fig. S9).

(c) Presence of macro-seeps (OS). The OS area is described in the Supplement (Section S3.2).

The three factors, faults, seismicity and presence of seeps, were applied on the gridded EMA as described in Section 6.4.

### 6.3 Attribution of the $\delta^{13}C$ value

Measured and published data of $\delta^{13}C$-$CH_4$ in gas MS are scarce and available only for a few petroleum fields. However, during the seepage process (migration driven by pressure gradients, i.e. advection), the $CH_4$ isotopic composition does not change significantly, so that surface $CH_4$ flux has basically the same $\delta^{13}C$ value of the original gas in the reservoir (e.g., Etiope et al. 2009). Therefore, for each field or basin, the MS

$\delta^{13}$C value was taken from published data related to subsurface reservoirs. A limitation of this strategy is that

in a given basin the MS gas may actually come from shallower reservoirs, not necessarily or not dominantly

from the deep productive reservoirs, which are more frequently the literature source of the isotopic value.

Therefore, in some cells the real isotopic value could be lighter than that used in the grid maps.

Accordingly we adopted the following procedure:

- when one or more seeps (OS) occur in a petroleum field (in the Petrodata list), the average $\delta^{13}$C-CH$_4$ of

those seeps was used for MS;

- in absence of seeps, reservoir $\delta^{13}$C-CH$_4$ data were used; they were taken from the inventory described by

Sherwood et al. (2017) or published literature. For the fields (in the Petrodata list) whose $\delta^{13}$C-CH$_4$ value is

not reported either in Sherwood et al. (2017) or literature, a theoretical $\delta^{13}$C-CH$_4$ value was estimated on the

basis of the type of gas (microbial or thermogenic) and maturity of the petroleum system.

The file contains 349 $\delta^{13}$C-CH$_4$ data points (from 891 petroleum fields). It was not possible to estimate a

specific $\delta^{13}$C value for the remaining 542 petroleum fields. In these cases, the global emission-weighted

isotopic value was used in the resulting empty cells, as described in Section 6.4.

### 6.4 MS gridding

PFA and OSA (described in the Supplement) were intersected with a high-resolution (0.05°x0.05°) global

grid. The 0.05°x0.05° cell dimension corresponds to the maximum resolution that can be obtained using

ArcGIS software (the software cannot handle shapefiles > 2 Gbyte). The high-resolution gridding was used

to match, as much as possible, the PFA: gridded PFA is in fact 14,791,897 km$^2$, while the original PFA was

13,033,750 km$^2$. The high-resolution gridding also served to reduce the boundary effect, and thus the

overestimation of the areas with MS enhancing factors, i.e. faults, earthquake and seeps (the larger the

cells, the higher the probability that the cells include MS enhancing factors).

As discussed in Section 6.2.1, only 57% of PFA cells were considered to host MS. It was then necessary to

delete 43% of PFA cells. The cells were randomly deleted only among those that do not host MS enhancing

factors (faults, earthquakes and seeps), i.e. empty cells (which are 93% of total PFA). The overall PFA

reduction of 43% was obtained by deleting 54% of the empty cells (resulting in a PFA of 8,408,360 km$^2$).

Combining PFA and OSA results in EMA (Table S6). The sequence of MS modelling is summarized in the

block diagram of Fig. S10. The MS levels were then assigned to the 0.05°x0.05° gridded EMA according to

the presence of the factors influencing MS: a) presence of faults; b) presence of seismic activity; c) presence

of macro-seeps (OS), as follows:

Level 1 was applied to cells without any geological factor.

Level 2 was applied to cells with faults or earthquakes

Level 3 was applied to cells with faults plus earthquakes or oil-seeps or gas-bearing springs

Level 4 is applied to cells with gas-seeps or mud volcanoes.

The resulting global MS CH$_4$ emissions are about 24 Tg yr$^{-1}$. The emissions per cell range from 14.7 tonnes

38  year$^{-1}$ (cells of about 30 km$^2$) to 29,446 tonnes year$^{-1}$ (cells of about 169 km$^2$). The grid was then converted

into 1°x1° resolution for atmospheric modelling applications (Fig. 7). MS emissions occur in 3,039 cells,
ranging from 15 to 471,000 tonnes yr$^{-1}$. The cell with the highest emission is located in the Caspian region
(Azerbaijan). The sensitivity of the MS modelling is discussed in the Supplement (Section S3.3).

*6.5 Evaluation of global MS emission and $\delta^{13}C$-$CH_4$*
The global MS emission derivable by summing the emission from the cells of the 4 MS classes, about 24 Tg
8 yr$^{-1}$ (Table S6), is within the range, 10-25 Tg yr$^{-1}$, previously suggested by Etiope and Klusman (2010). The
9 emission-weighted $\delta^{13}C$-$CH_4$ resulting from gridded MS is -51.4‰ (non-weighted average is -46.4‰). This
value is mostly influenced by areas with elevated MS of microbial gas, such as the Po Basin (Italy), the
Transylvania Basin (Romania) and the Powder River Basin (USA). The global emission-weighted value was
applied to cells without isotopic value.

*6.6 MS uncertainties*
*Spatial distribution uncertainty:* The uncertainty of the spatial distribution of MS depends on the assumption,
supported by field measurements, that MS occurs significantly only within petroleum fields (PFA) and areas
with seeps (OSA). The uncertainty of PFA depends on the "Petrodata" dataset of Päivi et al. (2007),
discussed in section 5.1, and it cannot be quantified. The uncertainty of OSA depends on the buffer applied
to individual seeps, which was however defined by geospatial analysis (see Supplement, S3.2).
*Emission uncertainty:* The uncertainty of the MS emission depends on the activity (EMA) and on the
process-based model of attribution of the seepage levels (emission factors), and their statistical elaboration,
discussed in section 6.2 (see also Fig. S10). Changing activity by ± 20% and emission factor by the 95%
confidence interval of the median, with different combinations, resulted in a total MS output ranging from 15
to 32.7 Tg yr$^{-1}$, with a mean of 23 Tg yr$^{-1}$, matching the first estimate (see Supplement). We can therefore
set, approximately, a maximum uncertainty in the total MS output of ±9 Tg yr$^{-1}$ (about ±38%).
The model was then tested comparing its output values with measured values. This comparison was
possible for 9 areas where the coordinates of the measurement points were identified. In all cases,
measured and modeled values have the same order of magnitude, and in many cases the range of the MS
level attributed by the model includes the mean value measured.
*$\delta^{13}C$ uncertainty:* The uncertainty of individual MS $\delta^{13}C$ values depends on the assumptions discussed in
Section 6.3. The uncertainty of the emission-weighted mean (-51.4‰) is mainly controlled by the cells with
larger MS emissions where $\delta^{13}C$ values are estimated. When the cells with emission-weighted mean are
excluded, the remaining 536 cells (at 0.05°x0.05°, over a total of 192,166) have emission ranging from 5623
to 8296 tonnes year$^{-1}$ and $\delta^{13}C$ values from -65 to -35‰ (mean -53.4‰). The difference of this value with the
emission-weighted mean, i.e., 2‰, may be considered as approximate expression of the uncertainty of the
global emission-weighted mean.

## 7. Geothermal manifestations (GM)

### 7.1 Global GM distribution

The global distribution of $CH_4$-emitting geothermal/volcanic sites (GM) generated here is based on an inventory of volcanoes and geothermal sites developed by Global Volcanism Program (2013) (see Sources of databases in the Supplement). This inventory reports all major volcanic-geothermal systems on Earth (2,378 sites; Fig. S12). They include both Holocene systems (1,307 sites distributed in 128 countries), and older, Pleistocene volcanic systems (1,071 sites distributed in 119 countries), which represent geothermal areas. In order to convert the point data into more realistic areal data (polygons), an arbitrary buffer area of 4 km of radius was created for each GM point (the buffer area does not influence the overall emission estimate, being only a parameter guiding the gridding). It is important to outline that this inventory reports the "zones" of volcanic/geothermal sites, and does not list individual manifestations: for example, the numerous geothermal manifestations in Central Italy are cumulatively included in a few lines, e.g., "Vulsini complex", "Sabatini complex" and "Vulture". Therefore each emission value, attributed as explained in Section 7.2, represents a "regional" GM emission.

### 7.2 Attribution of $CH_4$ emission levels

Methane flux measurements and regional total estimates in GM are available only in a few cases (<100 sites), mostly in Europe (as reviewed by Etiope et al. 2007). The GM inventory refers to geothermal-volcanic areas where GMs are expected to occur, but their actual surface area is unknown. Therefore, even assuming an emission factor (from the limited flux dataset) it cannot be translated into emission for each GM site. In this work, theoretical numbers were adopted considering 3 classes of "regional" emissions: 500, 5000, 10,000 tonnes $yr^{-1}$, as central values of the ranges 100-1000, 1000-10,000 and 5000-15,000 tonnes $yr^{-1}$, respectively. These ranges were derived from emission factors ranging from 1 to 150 tonnes $km^{-2}$ $yr^{-1}$ (Etiope et al. 2007) applied on a area of 100 $km^2$, as average order of magnitude of the extension of geothermal/volcanic zones (derived from Global Volcanism Program, 2013). Although the GM emission grid developed here is expected to improve global $CH_4$ inverse modeling (as it includes previously neglected GM sources), the total GM emission estimate suggested by the gridding, because of the uncertainty of the theoretical emissions, is not meant to update or refine the previous global GM emission estimate (derived by process-based modelling; Etiope, 2015).

The emission level was attributed based on:

(a) the location of the geothermal site, which may be within or outside a sedimentary basin (Fig. S13),

(b) the concentration of $CH_4$ measured in the geothermal fluids, within and outside a sedimentary basin.

The amount of methane in a geothermal-volcano area depends, in fact, on the presence of sediments rich in organic matter, which may be source of thermogenic gas in addition to the geothermal abiotic gas. The $CO_2/CH_4$ ratio of emissions to the atmosphere is in the order of 1000-10,000 in volcanic sites, with limited sedimentary contribution, and it ranges from 1 to 100 in geothermal systems characterized by important

sedimentary covers. Sediment-Hosted Geothermal Systems (SHGS) in sedimentary basins (e.g., Etiope, 2015; Procesi et al, submitted) show the highest $CH_4$ concentrations (lowest $CO_2/CH_4$ ratio). In addition, sedimentary basins hosting petroleum fields reasonably contain larger amounts of methane. The three classes of methane emissions are reported in the Supplementary Table S9.

### 7.3 Attribution of the $\delta^{13}C$ value

A specific dataset was compiled listing 98 published $\delta^{13}C$-$CH_4$ values of various, geographically dispersed, geothermal/volcanic systems in the world. The isotopic $\delta^{13}C$-$CH_4$ values range from -43.2 to -6.4‰, with an average of -26.7‰. The double-sided Grubbs test (Grubbs, 1969) identified 4 outliers; the mean $\delta^{13}C$-$CH_4$ value of the 94 values excluding the outliers is -26.5‰. It is known that geothermal methane in sedimentary basins, due to the presence of organic matter and related thermogenic gas, has a lower $\delta^{13}C$-$CH_4$ value compared to magmatic, sediment-free, systems (e.g., Welhan, 1988). The NPP of the $\delta^{13}C$-$CH_4$ data shows a sharp deviation at about -29‰ (Fig. S14). This value is actually consistent with the isotopic boundary of dominantly thermogenic gas; we used therefore this value as the limit between GM falling outside sedimentary basins and GM within sedimentary basins. The mean values of the two classes (excluding the outliers) are summarized in Table S8.

### 7.4 GM gridding

The GM shapefile generated in ArcGIS environment was spatially joined to the 1°x1° vector square grid. The result is reported in Table S9 and mapped in Fig. 8.

### 7.5 Evaluation of global GM emission and $\delta^{13}C$-$CH_4$

The 2,378 GM sites yield a total methane emission of about 5.7 Tg yr$^{-1}$, which is within the range of the latest global GS emission estimate (2.2-7.3 Tg yr$^{-1}$; Etiope, 2015). The emission-weighted mean value of $\delta^{13}C$-$CH_4$ for the GM emission is -30.6‰ (non-weighted mean is -27.5‰).

### 7.6 GM uncertainties

*Spatial distribution uncertainty:* The uncertainty of the spatial distribution of GM has the same uncertainty as the global distribution of geothermal-volcanic areas, derived from Global Volcanism Program (2013).

*Emission uncertainty:* The gridded GM emission, equivalent to the sum of individual regional values attributed as described in Section 7.2, has an uncertainty of about 75% (5.7±4.3 Tg yr$^{-1}$).

*$\delta^{13}C$ uncertainty:* The uncertainty of emission-weighted GM $\delta^{13}C$ may refer to the average of the two values corresponding to the 95% confidence interval of the means of the two groups of isotopic data (outside and within sedimentary basins) discussed in section 6.3, i.e. ±2.5‰.

3

## 8. Merging OS, SS, MS and GM: total geo-CH$_4$ emission gridding

### *8.1 Global geo-CH$_4$ emission*

The global geo-CH$_4$ emission distribution, obtained merging OS, SS, MS and GM grids, is shown in Fig. 9. The total gridded CH$_4$ emission is 37.4 Tg yr$^{-1}$ (Table 3, second column). The extrapolated gridded emission estimate including the factors not considered in the gridding procedure (i.e. mud volcano eruptions, existence of onshore and offshore seeps not included in the OS-SS inventories) is between about 43 and 50 Tg yr$^{-1}$ (Table 3, third column). These values are within the published global bottom-up estimates (Table 3, fourth column). The global extrapolated geo-CH$_4$ emission is then compatible with recent top-down estimates (about 50 Tg yr$^{-1}$ by Schwietzke et al. 2016; see also Section 9 for a wider discussion addressing the temporal variability of geological methane emissions). The scope of columns 3 and 4 in Table 3 (extrapolated and published emission estimates) is only to show that gridded emissions do not necessarily represent the actual global geo-CH$_4$ emission, because the datasets developed for the gridding may not be complete or may not contain the information necessary for improving previous estimates. Considering the four geo-CH$_4$ source categories individually, the gridded MS and GM emission totals are, however, within published ranges. The differences between gridded and published OS and SS are largely due to:
- incomplete OS dataset (it represents only 30% of global number of seeps assumed to exist on Earth)
- lower estimate of the global MV area (680 km$^2$ instead of 2800 km$^2$ assumed in previous works)
- incomplete SS flux dataset (flux data missing from at least 16 areas with known gas emissions).
The gridded emissions may represent an updated assessment of the global emissions only for MS and MVs (part of OS), because the gridding implied a careful assessment of the spatial distribution and emission factors for these types of geo-sources.

28

### 8.2 Global geo-CH$_4$ $\delta^{13}$C

Based on the emission-weighted $\delta^{13}$C value for each category of emission (using the respective emissions from Table 3), the global geo-CH$_4$ emission-weighted average $\delta^{13}$C is -49.4‰, considering global emission estimates and -48.5‰ for gridded emissions (Table 4). The global distribution of the isotopic signature is shown in Fig. 10.

35
36

### *8.3 Uncertainties of gridded geo-CH$_4$ distribution, emission and isotopic value*

The overall uncertainties of the spatial distribution of the geo-$CH_4$ sources, $CH_4$ emissions and emission-weighted average values of $\delta^{13}C$, depend on individual uncertainties of the four categories of seepage, as discussed in the respective Sections 4.6, 5.6, 6.6 and 7.6. These are summarized in Table 5.

**9 Note on temporal variability of geological methane emissions**

The fluxes of natural gas seepage from the Earth's crust are not constant, either on short (hours, days, months, seasons) or long (years, centuries, millennia) time scales. Seepage variations can be induced by endogenous (geological) and exogenous (atmospheric) factors, including subsurface gas pressure variations (controlled mainly by gas migration and accumulation processes), changes of fracture permeability (tectonic stress, seismicity), hydrostatic aquifer variations, meteorological and climatic changes (atmospheric pressure, temperature, humidity and microbiological activity in the soil; Etiope, 2015). Mud volcano episodic eruptions (Mazzini and Etiope, 2017), seismicity-related degassing (e.g., Manga et al. 2009) and seasonal variability of microseepage (higher in winter due to lower methanotrophic consumption in the soil; Etiope and Klusman, 2010), are three, well studied, examples of geo-$CH_4$ emission variability. Anthropogenic activity, through modification of aquifer pressures (water pumping) and petroleum exploitation (with consequent decrease of reservoir pressures) can also induce seepage variability over time (e.g., Etiope, 2015). Therefore the global geo-$CH_4$ emission reported in this work, as well as in all other estimates available in the literature, must be interpreted as average, present-day degassing. Substantial decadal changes of seepage could occur as a result of decadal changes of hydrostatic aquifer pressure (e.g., Famiglietti, 2014) and decadal changes of seismicity (e.g., Mogi, 1979). Specific empirical studies are however missing, and with the present state of knowledge it is impossible to provide a temporal variability factor.

On longer, geological time scales, a series of proxies suggested that geo-$CH_4$ emissions could have been quite variable over the Quaternary period (Etiope et al. 2008b). Recent estimates on geo-$CH_4$ emission at the end of Pleistocene deserve a specific discussion. Based on radiocarbon ($^{14}C$) measurements in methane trapped in ice cores in Antarctica, Petrenko et al. (2017) estimated the absolute amount of $^{14}C$-containing $CH_4$ in the atmosphere 11-12 k years ago, between the Younger Dryas and Preboreal intervals; this allowed to estimate that the maximum global natural, geological ($^{14}C$-free) $CH_4$ emission for that period was at most 15.4 Tg yr$^{-1}$. More recent analyses by the same authors confirmed this value (Dyonisius et al. 2018). These authors have then assumed that past geological methane emissions were no lower than today. They concluded, therefore, that present-day geological $CH_4$ emissions are much lower than present-day bottom-up estimates (54-60 Tg $CH_4$ yr$^{-1}$; Etiope 2015; Ciais et al., 2013). Without entering discussions on the accuracy and meaning of the ice core $^{14}C$-based analyses and their temporal extrapolation to today, the following investigates whether the estimate by Petrenko et al. (2017) is compatible with:

(a) the estimates provided by authors other than Etiope (2015) and those reported in the present gridding work,

(b) the lowest bottom-up geo-CH$_4$ emission estimates available so far,

(c) present-day top-down geo-CH$_4$ emission estimates derived by different techniques, and

(d) pre-industrial geo-CH$_4$ emission estimates based on ice-core ethane measurements and observed geo-CH$_4$-to-ethane ratios.

Table 6 summarizes the data including individual literature references. In the bottom-up estimates table, the third column reports the lowest estimates proposed on the basis of more recent datasets and emission factors, which are updated in comparison with the earlier estimates (reported in the second column). The last column reports the overall lowest estimates, from old and new works, i.e. the minimum emission values derivable from different extrapolations. This comparison shows that the Petrenko et al. (2017) estimate is lower than any bottom-up estimate, regardless of authorship. The top-down estimates table reports geo-CH$_4$ emission derivable by three different procedures:

(a) assessing the portion of $^{14}$C-free CH$_4$ in present day atmosphere (=30%; Lassey et al., 2007), then calculating the equivalent $^{14}$C-free CH$_4$ emission (30% of total CH$_4$ emission, ~558 Tg yr$^{-1}$ (Saunois et al. 2016) = 167 Tg yr$^{-1}$) and subtracting the anthropogenic $^{14}$C-free component (fossil fuel fugitive emissions from inventories ~ 100-130 Tg yr$^{-1}$; EDGARv4.2; Saunois et al., 2016). The natural component (geo-CH$_4$ emission) would be 37-67 Tg yr$^{-1}$.

(b) Using methane concentration and isotopic data from ice-core records, based on box modelling by Schwietzke et al. (2016), suggest a geo-CH$_4$ emission of 30-70 (50) Tg yr$^{-1}$.

(c) With the same box model plus 3D forward modeling, but using current day atmospheric methane and isotopic data, Schwietzke et al. (2016) suggested a current day total fossil fuel (oil/gas/coal industries plus geological) CH$_4$ emission of 150–200 Tg yr$^{-1}$. Considering that oil/gas/coal emission inventories indicate 100-130 Tg yr$^{-1}$, geo-CH$_4$ emission could be 20–100 Tg yr$^{-1}$, consistent with approach (b) but with a wide uncertainty range.

(d) Using ethane concentration data from ice-core records, the 3 Tg yr$^{-1}$ ethane top-down estimates by Dalsoren et al. (2018) confirm earlier bottom-up estimates of 2-4 Tg yr$^{-1}$ ethane (Etiope and Ciccioli, 2009). Observed geo-CH$_4$-to-ethane emission ratios would then suggest 42-64 Tg CH$_4$ yr$^{-1}$.

Overall, geo-CH$_4$ emissions derived by top-down estimates range between 20 and 100 Tg yr$^{-1}$. These values are consistent with bottom-up estimates but substantially higher than Petrenko et al. (2017) estimate. The following options should then be considered:

(i) All current-day bottom-up and top-down geo-CH$_4$ emission estimates are biased high.

(ii) The Petrenko et al. (2017) estimate is biased low.

(iii) All estimates are reasonable, but the assumption that past Younger Dryas to Preboreal geo-CH$_4$ emissions were not lower than today does not hold.

**10. Summary and Conclusions**

Gridded maps of global geological CH$_4$ emissions at 1°x1° resolution have been developed comprehensively for the first time for atmospheric modelling and evaluation of global CH$_4$ sources. The maps, elaborated by ArcGIS and provided as csv files, include the four main categories of natural geological CH$_4$ emissions:

onshore hydrocarbon seeps (OS), submarine (offshore) seepage (SS), diffuse microseepage (MS), and geothermal manifestations (GM). A combination of published and originally *ad-hoc* developed datasets was used to determine the emission factors and the areal distribution and extent (activity) of the several geo-$CH_4$ sources and their stable carbon isotope signature ($\delta^{13}C$). Due to the limited number of direct $CH_4$ flux measurements, globally and regionally representative $CH_4$ emission factors for OS, MS and GM were estimated based on experimental emission factors (measurements) and statistical approaches. Methane emission estimates for SS were adopted directly from published regional emission estimates. The results of this work can be summarized as follows:

(a) The global geo-$CH_4$ source map reveals that the regions with the highest $CH_4$ emissions are all located in the northern hemisphere, in North America, in the Caspian region, Europe, and in the East Siberian Arctic Shelf.

(b) The globally gridded $CH_4$ emission estimate (37.4 ± 17.6 Tg $yr^{-1}$ exclusively based on data and modelling specifically targeted for gridding, and 43-50 Tg $yr^{-1}$ when extrapolated to also account for onshore and submarine seeps with no location specific measurements available) is compatible with published ranges derived by top-down and bottom-up procedures.

(c) The procedures adopted to attribute $CH_4$ fluxes to mud volcanoes (MV, a OS sub-class) and microseepage (MS) are based on a detailed assessment of the activity (areas) and emission factors, and the resulting gridded total output can be considered a refinement of previously published emission estimates. Specifically, the global MV emission estimate (2.8 Tg $yr^{-1}$, excluding eruptions) is compatible with early estimates by Dimitrov (2002), Milkov et al (2003), Etiope and Milkov (2004) and Etiope et al (2008). Global MS emissions (previously estimated between 10 and 25 Tg $yr^{-1}$; Etiope and Klusman, 2010) are now estimated to be ~24 (±9) Tg $yr^{-1}$.

(d) Regional emissions of SS are available from the literature for only a limited number of cases. The regions with missing emission data in the literature are not included in the gridded dataset developed here. As a result, the gridded $CH_4$ emission estimate (3.9 Tg $yr^{-1}$) is substantially smaller than a previously published global total estimate (20 Tg $yr^{-1}$, which would include extrapolated values to regions without region-specific estimates (Kvenvolden et al. 2001). However, the published SS estimate has large uncertainties (at least 10 Tg $CH_4$ $yr^{-1}$ since two separate estimates of 10 and 30 Tg $CH_4$ $yr^{-1}$ were actually provided without indication of their uncertainties) and it was purely based on process-based modelling (Kvenvolden et al. 2001). This work verified that SS emissions also occur in other regions where emission values are missing (among these, the Gulf of Mexico, Caspian Sea and the North US Atlantic margin). Given an estimated SS emission factor, we propose that global SS $CH_4$ emissions may range between 5–12 Tg $yr^{-1}$, with a best guess (central value) of 8.5 Tg $yr^{-1}$.

(e) The emission-weighted global mean of $\delta^{13}C$-$CH_4$ is -48.5‰ for the gridded emissions, and -49.4‰ when gridded OS and SS emissions are extrapolated to include all global regions. The second value is therefore more realistic. Both values are significantly lower (about 4-5‰ lighter) than typical values attributed to fossil

fuel industry sources (-44‰ by Schwietzke et al, 2016) and much lower (10-11‰ lighter) than seepage values considered in inverse studies (-38‰ by Sapart et al. 2012). Clearly, natural geological sources are more [13]C-depleted than generally assumed (and this mostly occurs as microseepage and submarine seepage). Low maturity thermogenic gas and microbial gas are, in fact, a neglected, but considerable, fraction of the global fossil $CH_4$ budget (Sherwood et al. 2017). It is expected that using the updated, more [13]C-depleted, isotopic signatures in atmospheric modelling studies will increase the top-down estimate of the global geological $CH_4$ sources (all else equal).

The maps developed here represent important inputs for future atmospheric modelling of the global $CH_4$ cycle. Fossil fuel industry "upstream" activities (exploration, production, and some processing of fossil fuels) and associated $CH_4$ emissions occur largely on land surface above sedimentary basins that are also the habitat for geological $CH_4$ seepage. Thus, there is substantial spatial overlap in $CH_4$ emissions from the fossil fuel industry and geological seepage. Nevertheless, there is substantial spatial variability in $CH_4$ emission intensity for both the fossil fuel industry (Maasakkers et al. 2016; JRC/PBL, 2017) and geological seepage (this work). In the absence of a comprehensive gridded geological $CH_4$ seepage product, global or regional inverse model studies would erroneously attribute a low-bias to $CH_4$ emissions from geological seepage. This is because of a de-facto zero geological a priori estimate. At the same time, the inverse studies would erroneously attribute a high-bias to $CH_4$ emissions from fossil fuel industry activity (and potentially other sources) while correctly reporting total emissions of all sources. The geological seepage data and maps developed here can be used to refine fossil fuel industry and microbial $CH_4$ emission budgets at the regional and global level. Finally, methane/ethane and methane/propane source composition ratios are available for the four categories of geo-sources (preliminary data were used in Etiope and Ciccioli, 2009) for use beyond the scope of this work. Combining the gridded geo-$CH_4$ emissions and the available source composition data, gridded ethane and propane maps could be developed in the future. The gridded geo-$CH_4$ maps shall be updated when additional, statistically significant gas flux data for the several seepage categories become available. Geo-$CH_4$ emission from a fifth, recently discovered, geological category, the seepage from serpentinized peridotites (e.g., Etiope and Schoell, 2014; Etiope et al. 2017 and references therein) shall also be gridded when sufficient flux data become available.

**11 Data availability**

The free availability of the data does not constitute permission for their publication. If the data are essential to new modelling, or to develop results and conclusions of a paper, co-authorship may need to be considered. Grid csv files (emission and isotopic composition for each geological source and integrated grid files) and microseepage and geothermal manifestations inventories are available at https://doi.org/10.25925/4j3f-he27 (Etiope et al. 2018), including full contact details and information on how to cite the data. The SS inventory is provided in the Supplement (Table S4). Due to CGG (2015) license restrictions, the OS inventory can be requested at

*www.cgg.com/en/What-We-Do/Multi-Client-Data/Geological/Robertson-Geochemistry.*

The datasets of petroleum fields, faults, volcanic-geothermal sites, earthquakes, sedimentary basins are available on the web as described in the Supplement.

The Supplement related to this article is available online at ………….

Competing interests. The authors declare that they have no conflict of interest.

**Acknowledgements.** The work was supported by NASA grant NNX17AK20G. Thanks are due to Lori Bruhwiler for revising the manuscript.

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

**Tables**

Table 1. Parameters and data sources used to generate grid maps of geological $CH_4$ sources. Complete references and links to data sources are provided in the Supplement.

| | Onshore seeps (OS) | Submarine seeps (SS) | Microseepage (MS) | Geothermal manifestations (GM) |
|---|---|---|---|---|
| **Activity data** | Global seep distribution (georeferenced points) | Global distribution of marine seepage zones (georeferenced areas) | Global distribution of petroleum fields (georeferenced area) | Global distribution of volcanoes and geothermal sites (georeferenced points) |
| *Data source* | Updated GLOGOS dataset (after CGG, 2015; Etiope, 2015) | Originally developed data-set | "Petrodata" from Päivi et al. (2007) | Global Volcanism Program (2013) |
| **Emission factors** | Measurements and estimates based on size and activity | Measurements and estimates based on size, activity and depth | - Statistical evaluation of flux data<br>- presence of faults<br>- seismicity | - Measurements and estimates based on size and activity<br>- presence of sediments |
| *Data source* | Literature, web sources | Literature | Merged global and regional databases<br><br>USGS Earthquake Lists, Maps and Statistics | Literature<br><br>Sedimentary basins world map (CGG data services) |
| $\delta^{13}$C-CH$_4$ | Measured or estimated value for each seep | Mean value for each seepage zone | Mean value for each basin or sub-basin | Global mean value based on statistical analysis |
| *Data source* | Updated GLOGOS (CGG, 2015), reservoir data (Sherwood et al. 2017) and petroleum system data (literature) | Published data or estimates based on local petroleum system | Petroleum reservoir data (Sherwood et al. 2017 and literature), seeps (OS data-set) and estimates based on the type of petroleum system | Literature data and estimates based on the type of system |

Table 2. Estimates of global $CH_4$ emission from OS (onshore MV and other seeps)

| | MV quiescent degassing | MV quiescent + eruption | Gas-oil seeps | Total quiescent | Total (incl. MV eruptions) |
|---|---|---|---|---|---|
| Dimitrov (2002) | 0.3 – 2.6 | 10 - 12 | nd | nd | nd |
| Dimitrov (2003) [a] | < 2.3 | < 5 | nd | nd | nd |
| Milkov et al (2003) [a] | < 2.9 | < 6 | nd | nd | nd |
| Etiope and Milkov (2004) | 2.8 - 4 | 5.6 - 8 | nd | nd | nd |
| Etiope et al (2008) [a] | < 3 - 4.5 | < 6 - 9 | 3-4 | 6-8.5 | 9-13 |
| Etiope et al (2011) | 9 | < 10-20 | 3-4 | 12-13 | 13-24 |
| This work – 2827 seeps | 2.83 | nd | 1 | **3.8** | nd |
| This work – total extrapol. | ~3 | 6.1 | ~ 2 | ~ 5 | ~ 8.1 |

nd: not determined

[a] Values include shallow submarine MV, therefore they can be considered as upper limits for onshore emission.

2
3 Table 3. Global gridded, global extrapolated and global published geo-CH₄ emissions

| Emission category | CH$_4$ gridded emission (Tg yr$^{-1}$) | CH$_4$ extrapolated * emission (Tg yr$^{-1}$) | Published ranges (best guess) (Tg yr$^{-1}$) |
|---|---|---|---|
| OS - Onshore Seeps | 3.8 [a, b] | 8.1 | 9 – 24 [d] |
| SS - Submarine Seeps | 3.9 [c] | >7 | 10 - 30 (20) [e] |
| MS - Microseepage | 24 | 24 | 10 – 25 [f] |
| GM - Geothermal Manifestations | 5.7 | 5.7 | 2.2 - 7.3 [g] |
| Total | 37.4 [a, b, c] | 42.8 – 49.8 | 41- 76 (58) |

* Including estimates from notes a, b, and c. See also text below. [a] Not including MV eruptions
[b] Partial (estimated <50%) gas-oil seeps emissions. [c] Excluding unidentified or not-investigated offshore seepage sites
[d] Etiope et al. 2008; Etiope et al. 2011 (see also Table 3). [e] Kvenvolden et al (2001)
[f] Etiope and Klusman (2010). [g] Etiope (2015)

Table 4. Global emission-weighted $\delta^{13}C$ values (‰)

| Emission category | Emission-weighted $\delta^{13}C$ |
|---|---|
| OS - Onshore Seeps | -46.6 |
| SS - Submarine Seeps | -59 |
| MS - Microseepage | -51.4 |
| GM - Geothermal Manifestations | -30.6 |
| Global weighted average (based on gridded emissions, 2$^{nd}$ column in Table 3) | **-48.5** |
| Global weighted average (based on globally extrapolated gridded emissions, 3$^{rd}$ column in Table 3) | **-49.4** |
| Global weighted average (based on published emissions, 4$^{th}$ column in Table 3) | **-49.8** |

15 Table 5. Summary of uncertainty factors for the four types of gridded geological emissions

| Emission category /Uncertainty | Spatial distribution | Emission | $\delta^{13}C$* |
|---|---|---|---|
| OS - Onshore Seeps | Zero uncertainty on global scale<br>Coverage 30% gas-oil seeps (but all biggest seeps reported)<br>Almost complete MV coverage | Gas-oil seeps uncertainty: max. 90% (order of magnitude theoretically assessed)<br>MV uncertainty: 48% (statistically assessed values)<br><br>Overall OS uncertainty 58% (±2.2 Tg yr$^{-1}$) | ±1‰ |
| SS - Submarine Seepage | Zero uncertainty for central values of gridded area<br>Area extent from published papers<br>Unknown % of global coverage (likely >80% ?) | From published data (central value used)<br><br>Uncertainty 54% (±2.1 Tg yr$^{-1}$) | ±7‰ |
| MS - Microseepage | Theoretically predicted (measurements and process-based model)<br>Possibility that microseepage occurs outside petroleum fields (unknown gas pools) is accounted for | Process-based modelling<br><br>Uncertainty max. 38% (< ±9 Tg yr$^{-1}$) | ± 2 ‰ |
| GM - Geothermal Manifestations | Zero uncertainty | Process-based modelling (regional emissions)<br><br>Uncertainty 75% (±4.3 Tg yr$^{-1}$) | ±2.5‰ |

17 * Uncertainty of the global emission-weighted $\delta^{13}C$ values of Table 4.

2
3

5 Table 6. Combinations of bottom-up and top-down estimates of geological methane emissions (Tg yr$^{-1}$)

7 Bottom-up

| | Lowest estimates from other authors | Lowest updated estimates | Lowest overall estimates |
|---|---|---|---|
| Onshore macro-seeps (includ. mud volcanoes) | 5[a] | 3.8[e] | 3.8[e] |
| Global submarine emissions | 10[b] | 5[e] | 5[e] |
| Global microseepage | 7[c] | 10[f] | 7[c] |
| Geothermal | 5.5[d] | 2.2[g] | 2.2[g] |
| Total | 28 | 21 | 18 |

8 Top-down

| | |
|---|---|
| Atmoshere $^{14}$C-based (Etiope et al. 2008; Lassey et al., 2007) | 37-67 |
| Ice-core $^{13}$C-CH$_4$ based (Schwietzke et al., 2016) | 30-70 |
| Current day emission data (Schwietzke et al., 2016) | 20-100 |
| Ice-core ethane (Dalsoren et al., 2018) and observed geo-CH$_4$-to-ethane ratios (Etiope et al., 20009) | 42-64 |

[a] Dimitrov (2003); [b] Kvenvolden et al. (2001); [c] Klusman (1998); [d] Lacroix (1993); [e] this work; [f] Etiope and Klusman (2010); [g] Etiope
(2015).

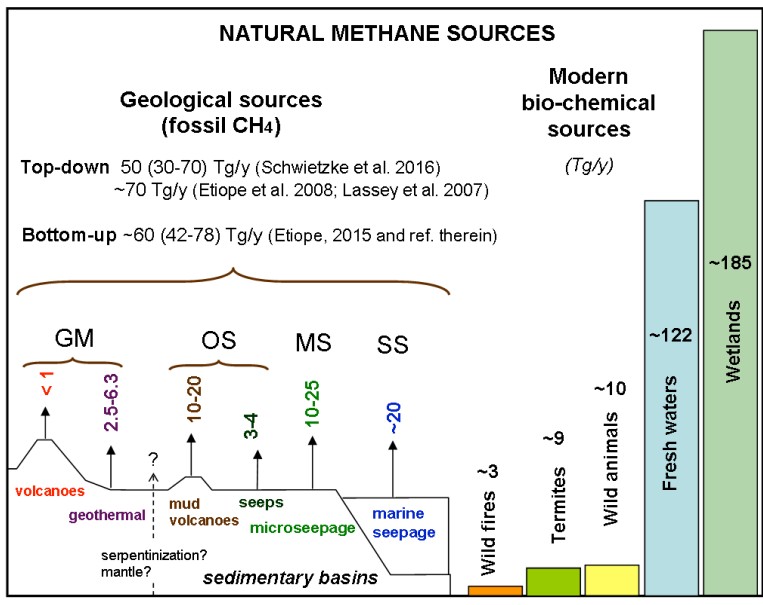

Fig. 1. Sketch of geo-$CH_4$ sources, their global emission estimates (after Etiope, 2012 and Etiope, 2015) and comparison with other natural $CH_4$ sources (bottom-up estimates from Saunois et al. 2016). GM: Geothermal Manifestations, OS: Onshore Seeps, MS: Microseepage and SS: Submarine Seepage.

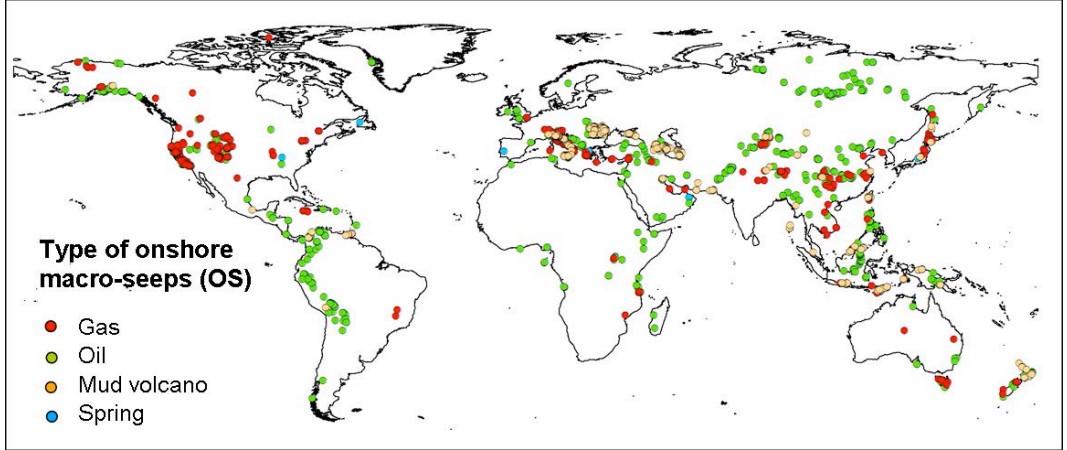

Fig. 2 Global distribution of onshore seeps (OS)

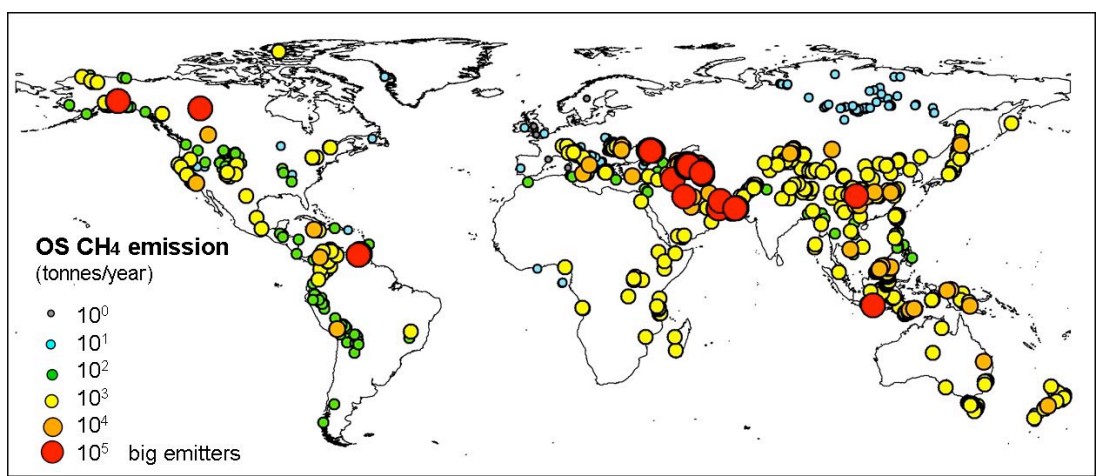

Fig. 3 Distribution of the order of magnitude of methane emission from onshore seeps.

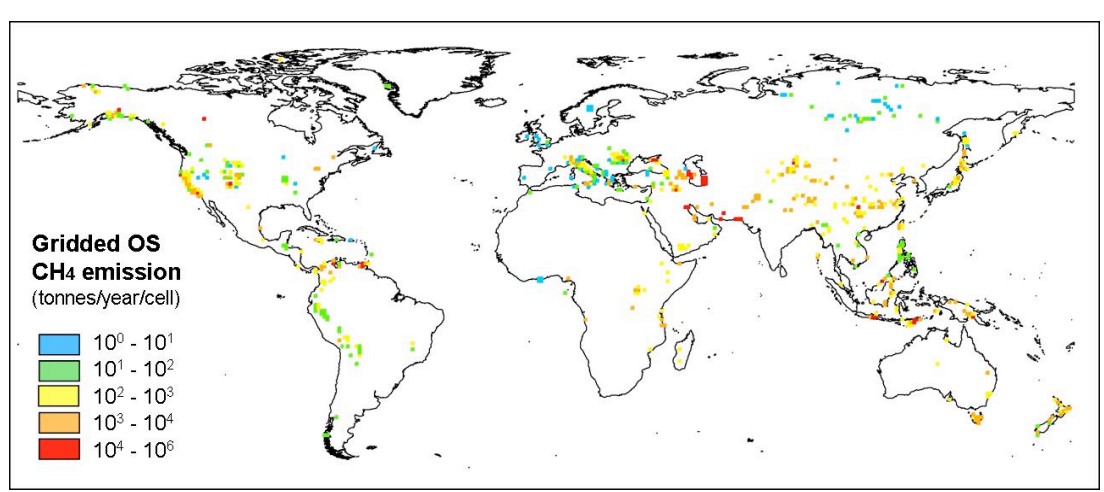

Fig. 4  Gridded map of OS methane emission. This map refers to the csv file "OS_output_2018"

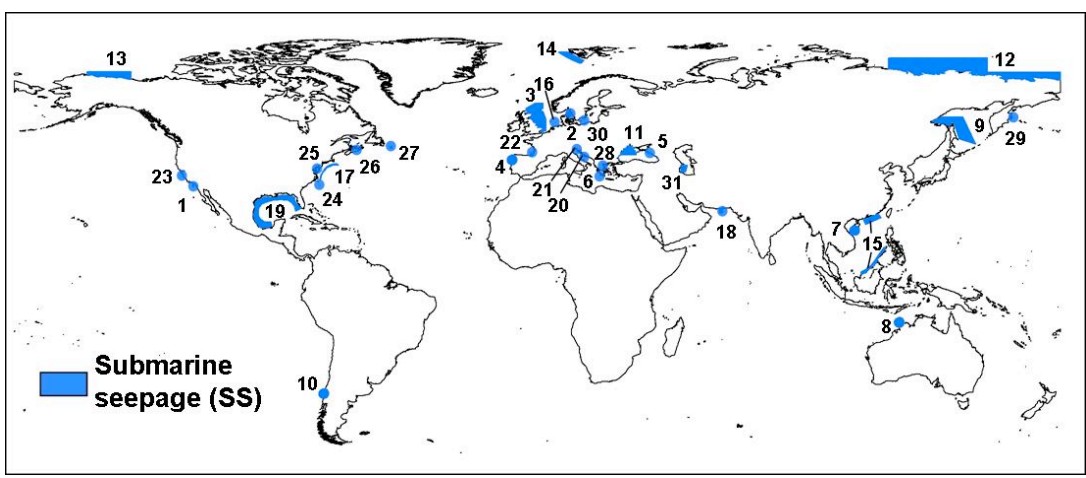

Fig. 5 Distribution of submarine seepage (SS) areas. SS numbering refers to the list in Table S4
(circle symbols mark small areas sites)

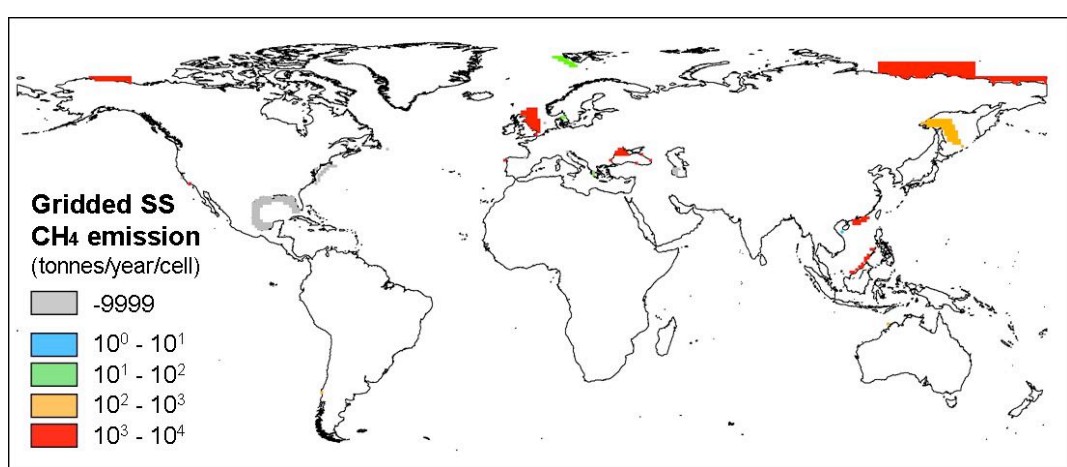

Fig. 6  Gridded map of SS methane emissions. This map refers to the csv file "SS_output"_2018.

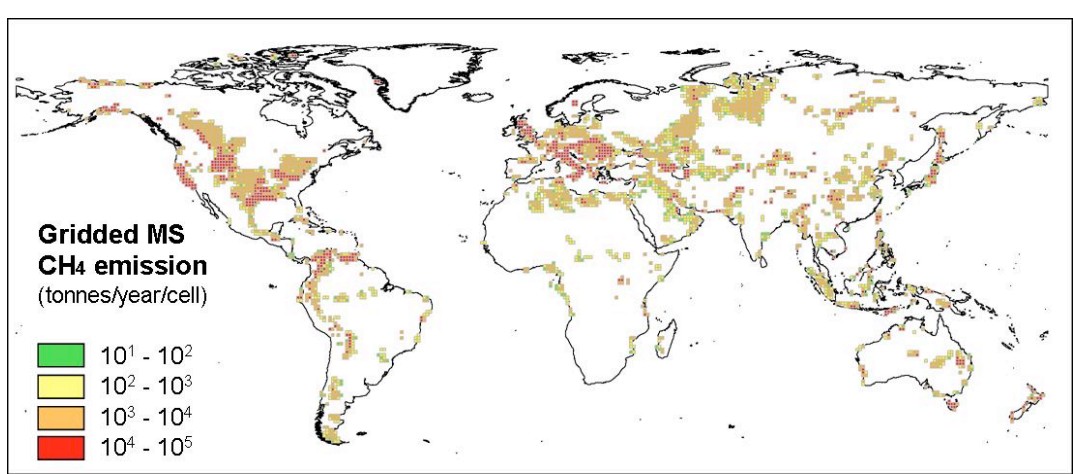

Fig. 7 Gridded map of MS methane emission. This map refers to the csv file "MS_output_2018"

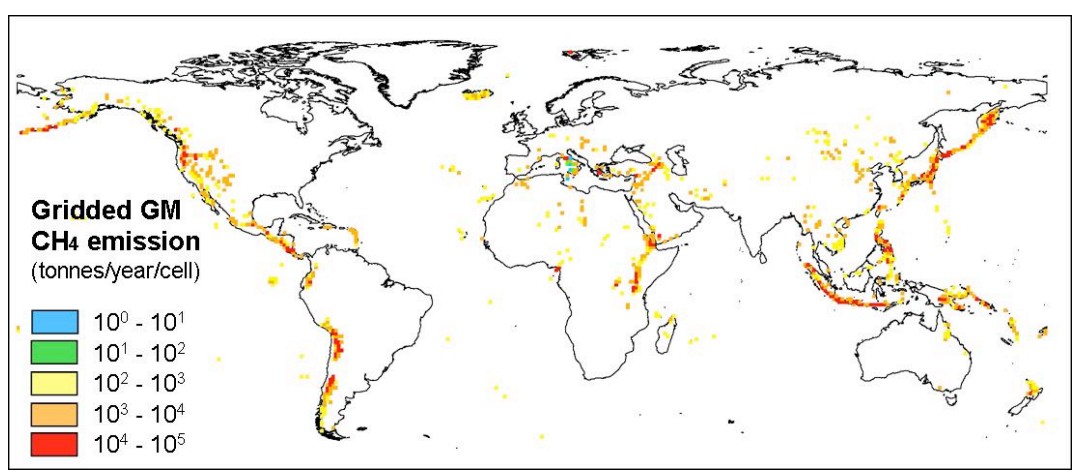

Fig. 8  Gridded map of GM methane emission. This map refers to the csv file "GM_output_2018"

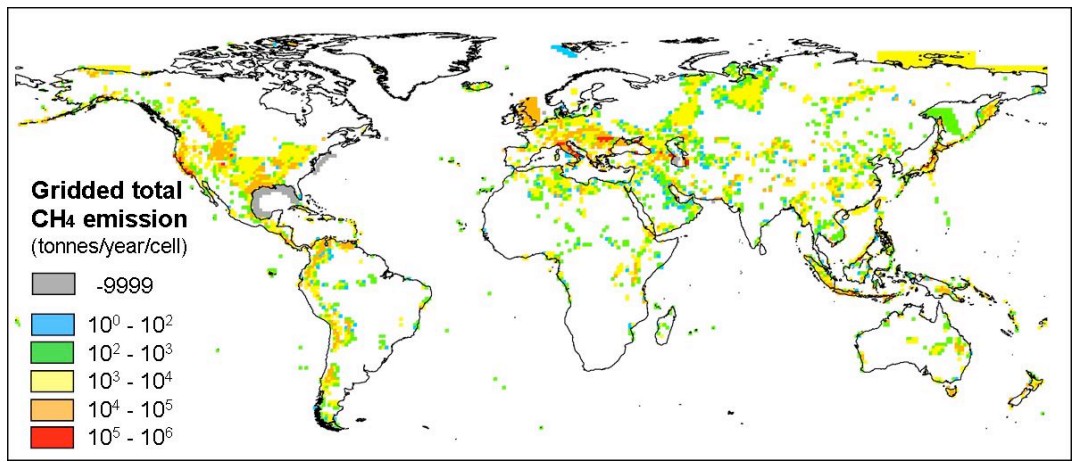

Fig. 9  Gridded map of total methane emission from OS+SS+MS+GM.
This map refers to the csv file "Total geoCH4_ output_2018".

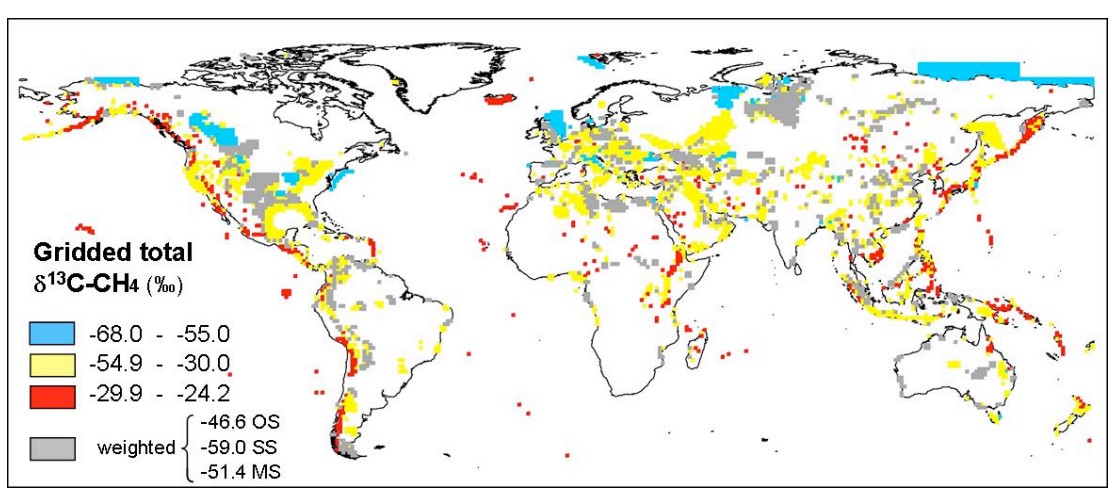

Fig. 10  Gridded map of integrated $\delta^{13}$C values of OS+SS+MS+GM (emission-weighted within each category).
"Weighted" (grey) refers to OS and MS sites where the weighted $\delta^{13}$C value (Table 4) is used replacing -9999.
This map refers to the csv file "Total geoCH4_13C_2018".