# Peer review of "Gridded maps of geological methane emissions and their isotopic signature"

_Earth System Science Data, 2018_

## Referee Comment (RC1) · Anonymous Referee #1 · 22 Oct 2018

Gridded maps of geological methane emissions and their isotopic signature Giuseppe Etiope et al.

This paper attempts to fulfil the very real need for a global evaluation of geological sources of atmospheric methane – a generally understated and commonly overlooked source of this important Greenhouse Gas. The authors have undertaken the daunting task of summarising the methane emissions in a logical way – subdividing the planet into a 1° x 1° grid, and populating this grid with the data from four logically-selected categories of natural geological methane source. Available data from each grid square are summarised by two values for each category: emission strength and carbon isotope – again, a logical approach. The data set is thus a valuable resource for those interested in global Greenhouse Gas emissions for modelling, or whatever purpose.

[Figure]

Inevitably there are shortcomings to this approach: • the number of source references (published papers, reports etc.) is vast: most, but not all, have been utilised. Table S4, for example, is not complete. • there is an enormous volume of data acquired by, for example, the petroleum industry, which has not been released to the public domain, so the data set cannot be comprehensive. • such data sets are never complete. As soon as a compilation is completed, more source references appear. For the above reasons it would perhaps be appropriate to recognise this a 'provisional' attempt to evaluate global geological methane emissions. However, I believe it be the only one available, and in its present form it is more than adequate to demonstrate the significance of geological sources of methane.

Are there plans to maintain and update the data set?

These comments do not detract from the value and usefulness of this data set. It is presented in a suitable format for use on GIS systems etc., and the paper adequately describes how the data set was generated. To the best of my knowledge the data set is unique, compiled in an appropriate manner from data of suitable quality.

Rating: 2. It falls short of 1 only because of the shortcomings identified above.

Specific comments: p3 l20: four major categories: is it worth noting here that other sources (e.g. deep water seeps) do occur, but are less likely to be responsible for direct methane emissions to the atmosphere?

P3 l23: Submarine (offshore) seeps: presumably this includes offshore mud volcanoes; if so this should be stated - if not they should have been included either here or in a separate category.

P3 l26: diffuse microseepage: presumably this category is exclusively onshore - this should be stated.

P4 l 24 onshore cells without OS, GM or MS sources?

P16 l24 The double-sided Grubs test should either be explained or a suitable reference

should be provided.

P16 l28 "as limit" – or "as the limit"?

P25 l 14 (Table 5). The distribution of submarine seepages: "unknown % of global coverage (likely >80% ?)". What does this mean? If this means that more than 80% of the global distribution of submarine seeps is accounted for in the data set, I strongly dispute this. I suspect that many more than 20% of existing seeps and seep areas remain undiscovered (or are discovered but unreported).

The Supplement is first mentioned on p4 l8 –instructions on how to access it should be provided here and on p20 l20.

Technical comments: General: • standardise "per": e.g. "Tg year-1" OR "Tg/y". • p3 l11 "ad hoc" is correctly in italics. "et al." should also be in italics. • Several occurrences of "et al" should be corrected to "et al."

p2 l30: Bergamaschi et al. 2014: listed as 2014 in References

p3 l32: Etiope et al. (2007) not in References

p6 l 26-7: "OS emissions in the order of . . .. " Clumsy wording. Suggest: "There is a total of 76 OS with emissions in the order of 104 t CH4 year-1 ....."

p9 l1: California is in the USA!

p11 l10-11: Klusman et al. 2008 – not in References.

p11 l 28 "Sciarra": written "Sciarpa" in References. Which is correct?

p15 l24: Global Volcanism Program (2013): details of this should be included in the References

p15 l38 AND p16 l 6 Etiope et al (2007) not in References.

p16 l16 Procesi et al. – details should be added to References.

p17 l9 – there is a superfluous fullstop [".."] at the end of the line.

p19 l31 "JBC/PBL": should this be "JRC/PBL" as in References?

p22 l39 "breathdglobal" should be "breath global"

p22 l50 should Las Animas and Huerfano counties have capital leading letters?

The following appear in the References, but are not cited in the text: • Le Quéré et al., 2013 • Etiope, Baciu, Caracausi, Italiano & Cosma, 2004 • Etiope, Christodoulou et al. 2013 • Etiope Doezma & Pacheco, 2017 • Etiope, Feyzullaiev et al. 2004 • Etiope & Schoell, 2014 • Saunois et al. 2017 • USGS World Energy Assessment Team, 2000

I have not gone through the Supplementary References so I suggest that the authors re-check that it is correct and complete.

---

## Author Comment (AC1) · 22 Oct 2018

R: Reviewer A: Author

(. . .) R: Inevitably there are shortcomings to this approach: âËŸA 'c the number of source references (published papers, reports etc.) is vast: most, but not all, have been utilised. Table S4, for example, is not complete. âËŸA 'c there is an enormous volume of data acquired by, for example, the petroleum industry, which has not been released to the public domain, so the data set cannot be comprehensive. âËŸA 'c such data sets are never complete. As soon as a compilation is completed, more source references appear. For the above reasons it would perhaps be appropriate to recognise this a 'provisional' attempt to evaluate global geological methane emissions.

[Figure]

However, I believe it be the only one available, and in its present form it is more than adequate to demonstrate the significance of geological sources of methane.

A: We agree with Reviewer#1, Table 4 (Submarine Seepage) may not be complete as it can only refer to published data (as indicated in Section 5.1); we cannot include unpublished or confidential data from petroleum industry. We outline, however, that the data, for our purpose, must only refer to methane emission into the atmosphere, not to data on gas flux at seabed or to the existence of submarine seeps. We doubt that oil industry is interested in estimating the flux of methane at the sea surface. We can however better clarify the point by rephrasing the following sentence in Section 5.1: A specific dataset….(…...)…..was developed based exclusively on published literature (Table S4 in the Supplement).

R: Are there plans to maintain and update the data set?

A: Yes, the dataset can be updated annually.

R: These comments do not detract from the value and usefulness of this data set. It is presented in a suitable format for use on GIS systems etc., and the paper adequately describes how the data set was generated. To the best of my knowledge the data set is unique, compiled in an appropriate manner from data of suitable quality. Rating: 2. It falls short of 1 only because of the shortcomings identified above.

A: Ok, but as explained above, we can only report published data, so how can Table 4 be a shortcoming? There is no choice…..

Specific comments:

R: p3 l20: four major categories: is it worth noting here that other sources (e.g. deep water seeps) do occur, but are less likely to be responsible for direct methane emissions to the atmosphere?

A. In Section 5.1 we clarified that deep water seeps are not considered because of the limited or nil impact to the atmosphere
R: P3 l23: Submarine (offshore) seeps: presumably this includes offshore mud volcanoes; if so this should be stated - if not they should have been included either here or in a separate category.

A: Yes, they include offshore mud volcanoes

R: P3 l26: diffuse microseepage: presumably this category is exclusively onshore – this should be stated.

A: Yes, this will be clarified at the beginning of Section 6.1

R: P4 l 24 onshore cells without OS, GM or MS sources?

A: No, only, OS and GM; all onshore MS cells have a value

R: P16 l24 The double-sided Grubs test should either be explained or a reference should be provided.

A: OK reference is going to be provided

R: P16 l28 "as limit" – or "as the limit"? OK

R: P25 l 14 (Table 5). The distribution of submarine seepages: "unknown % of global coverage (likely >80% ?)". What does this mean? If this means that more than 80% of the global distribution of submarine seeps is accounted for in the data set, I strongly dispute this. I suspect that many more than 20% of existing seeps and seep areas remain undiscovered (or are discovered but unreported).

A: We outline again that we are referring exclusively to shallow seeps (<300-400 m below sea level), with impact in the atmosphere. We considered conservatively 80% with a question mark; would 60 or 70% be more realistic?

R: The Supplement is first mentioned on p4 l8 –instructions on how to access it should be provided here and on p20 l20.

A: Following the editorial rule, the access to the Supplement must be indicated at the

end of the manuscript. We did not know the doi during submission. It will be added in the revised final version.

A: We are going to address all minor technical comments.

We are very grateful to Reviewer#1 for the careful and positive review and comments.

---

## Referee Comment (RC2) · Anonymous Referee #2 · 23 Oct 2018

The manuscript presents a global map of geological methane emissions, accounting for on-shore, off-shore macro and micro seepage as well as geothermal emissions. To test the consistency between inventory estimates and atmospheric data it is crucial to have such a map. Until now, it was left up to atmospheric modelers to make their own maps using the available information, and make assumptions regarding missing inputs – such as globally representative isotopic fractionation factors. Clearly, the experts in geologic emissions are much better equipped for this task. Therefore, the effort that went into this study is serves a valuable purpose. With that an important requirement for publication is met: The manuscript fits well in the scope of the journal and will have scientific impact. The next important requirement is that the information that is provided has a solid scientific basis. This may be the case, but is much more difficult to judge.

[Figure]

As explained further below, the description of the method is very brief and it is not always clear which data are used, and where they come from. This makes it difficult to verify the numbers, let alone to reproduce the inventory from the underlying data sources. Revisions will be needed to improve transparency.

GENERAL COMMENTS

According to the abstract the global mean isotopic signature of geologic emissions is -48.5 to -49.4 per mil. In addition to the global average, a global map of fractionation factors is provided. The question is how those numbers were derived. The section on on-shore seeps mentions (section 4.3) that it is a combination of measurements and estimates. The measurements are 'as indicated in the literature' without references. The estimates follow 3 rules that are listed without specifying which rule applies when. A global mean is used in regions for which no data exist. It is unclear, however, where this applies. The descriptions of the other subsections on isotopic information are similarly general, and unspecific. It is unclear how the range between -48.5 and -49.4 should be interpreted. Is this supposed to be an uncertainty range? Some of the paragraphs dealing with uncertainty in d13C mention 15 per mil as a range of reported numbers. Then how is it possible to arrive at an uncertainty on the global fractionation within 1 per mil? These short comings will have to be repaired.

Important for the application of the geologic emission map to inverse modelling is the information on uncertainty that is provided. However, the method that is used to quantify uncertainties is questionable. For example, for some sources (e.g. OS and SS) it is stated that the uncertainty in the geographical distribution is practically zero. While this may apply to specific sources that have been located, there should be uncertainty from sources that have not yet been found. The question then becomes to what fraction of the emissions this may apply. Section 4.6 on onshore seeps describes contributions to uncertainty, but does not provide a single number. For the uncertainty in the isotopic fractionation factor a maximum difference between estimates is mentioned, but it is unclear how representative that maximum is. Table 5 mentions ±1 per mil, which seems
quite accurate, but it is unclear where this number comes from. In the description of GM uncertainties, the emission estimates are discussed without providing a clue of what the uncertainties may be. To summarize: The treatment of uncertainties requires several clarifications.

If I understand correctly the gridded emission maps do not account for the global extrapolation, i.e. they account for 37 Tg/yr of methane emissions. For the remainder no specific information is available, which raises the question what the extrapolation of almost a quarter of the emissions is based on. The assumptions underlying the extrapolation are not spelled out, which is critical not only for trying to understand how they were derived, but also to guide users of the emission dataset as to where missing emissions are to be expected (see my earlier remark about the uncertainty in the spatial distribution of the emissions). Besides a clearer description of the assumptions underlying the extrapolation, some discussion is needed of what are reasonable assumptions (in terms of geographical regions) to put the remaining sources or to account for their uncertainty.

Some discussion is needed of how to distribute the emissions not only spatially but also temporally. Without this information, modelers will probably assume that all emissions are constant over the year. Besides eruptions, for which you would obviously need to know the timing, continuous emissions may vary with environmental conditions (temperature, soil water content?). A few sentences of discussion would be useful to provide information to guide the choice of temporal distribution, and the uncertainty assigned to it (for atmospheric modelers it is relevant to know within what bounds continuous emission may vary over the year)

SPECIFIC COMMENTS

Page 3, line 10: What are 'originally ad hoc developed datasets'? Which parts of the inventory are based on such data?

Page 4, line 8: '... reported in the supplement'. Add '(S6)'.

Page 4, line 14: How were the 'single OS, SS, MS, and GS shapefiles' derived? Aren't these just lists of coordinates of reported seeps? If so, this needs to be made clear to avoid the impression that unspecified information went into these shape files.

Page 5, line 13: '... listed without coordinates' Does this mean that no geographical information is provided at all?

Page 5, line 15: 'The total number of 3439 OS represents about 30% ...' Does this mean that the remainder is part of the 'global extrapolation'? This question applies as well to the 50% mentioned later in this paragraph.

Page 5, line 33: 'theoretical values were used'. The explanation that follows makes clear what these values account for, but it is unclear what the values are and based on which criteria they are assigned.

Page 5, line 38: What is the difference between 'micro' and 'mini' seepage?

Page 6, line 38: As discussed on Etiope et al, 2009, the isotopic signature of the reservoir may be a poor indicator of the isotopic signature of the emissions due to fractionation due to advective segregation. In light of this, what is justifying the use of the reservoir signature here?

Page7, line 8: '... because of multiple counting of 57 seeps ...' but I thought the point of using ArcGIS was to deal with this kind of issues. Why not assign the emission proportional to area or something like that? There must have been an easy way to avoid double counting.

Page 7, line 16: '... OS grids are not meant to update or refine ...' but the reference is to a paper that was published 10 years ago. Table 1 lists data sources for category OS that are of more recent dates. What does this statement mean for those updates?

Page 7, line 22: '...those estimates are indicated in the Table as upper limits ...' Are marine MV larger emitters as onshore MV's? Otherwise I don't understand this statement. It seems not obvious to me, since a fraction of the emissions from submarine

MV's will be oxidized in the water column.

Page 8, line 10: The total mean value is just the average of all reported d13C values? Since emission weighting seems such an obvious improvement of this estimate why are both estimates mentioned?

Page 8, line 35: '< 500 m deep ... McGinnis et al, 2006' According to McGinnis it is unlikely that seeps from deeper than 100m can contribute significant amounts of methane to the atmosphere. Therefore, the threshold should be 100m instead of 500m.

Page 11, line 34: 'The similar order of magnitude ... log normal behavior' If the mean and median are the same than this suggests rather a normal distribution. I don't see why the mean and median in the same order would point to a log normal distribution.

Page 13, line 23: Implicit here is that the shallower reservoirs have a heavier isotopic signature than a deep reservoir. Is this generally true? If so, then why?

Page 14, line 2: What explains the ~10% difference between gridded and reported area in this case?

Page 14, line 37: 'buffer applied to individual seeps' I don't understand what this means. The reference should probably be to section 6.2.1.

Page 15, line 16: The result of 2 different ways of averaging does not sound as a reliable estimator of uncertainty. This seems confirmed by 2 per mil being very small given the range of the fractionation values.

Page 16, line 6: 'Accordingly, the total emission estimate ... (Etiope et al, 2008)' Why is this? Because the accounting approach that is adopted here is not considered meaningful?

Page 17, line 4: What explains the exceptionally heavy isotopic signature of GM emissions?

Page 18, line 2-5: If I understand correctly, the emission maps are referred to here as

gridded emissions. It means that they only account for 37 Tg/yr.

Page 18, line 10: The combined d13C uncertainty depends on the individual uncertainties. The more important question of how they are combined should also be answered.

Page 19, line 23-24: 'It is expected that using the updated . . . (all else equal)' I don't understand why this would be the case. At -49 per mil atmospheric 13C is really insensitive to geological emissions as it is so close to the mean atmospheric composition.

Page 19, line 32-34: It is unclear to me why this would be the case (see my previous point).

Suppl. Page 9: '. . . considered for the text file' Which text file?

Suppl. Fig S3: To what extend could the difference in slope between the two regression lines be explained by the use of the erroneous syringe method for the larger MV's (increasing the micro seepage would bring the lines closer together)

Suppl. Page 8: 'tested' or 'evaluated' i.o. 'checked'. The latter suggests that the validity of the sensitivity was verified using some external information, which is not the case.

TECHNICAL CORRECTIONS

Page 6, line 31: 'emission' i.o. 'output'

Page 8, line 37: 'emission' i.o. 'output'

Page 16, line 24: 'Grubbs' i.o. 'Grubs'

Table S3, caption: 'Azerbaijan' i.o. 'Azerbaiajn'

---

## Referee Comment (RC3) · Anonymous Referee #3 · 25 Oct 2018

Review of the paper by G. Etiope et al. entitled "Gridded maps of geological methane emissions and their isotopic signature"

General comments :

This paper proposes the first gridded map of (natural) geological methane emissions for four categories, together with an estimate of their isotopic signatures, and ratios methane/ethane and methane/propane. These products are of primary importance to help closing the global methane budget, and especially to be used as a prior description in atmospheric inversions. A very large amount of work has been done to produce this map, based on the long experience and recognized expertise of the first author and there is no doubt that these emissions gridded description will be widely used.

[Figure]

Balancing a bit this large interest for paper's products is the content of this discussion paper. The structure of the paper is satisfying but I have several general concerns and many specific ones listed below:

1/ Abstract and introduction needs attention (see specific comments)

2/ You quote 20 self publications (Etiope or Etiope et al). It seems a bit too much regarding the total number of references and I recommend to keep only the main ones. Also, some recent relevant references are missing such as Petrenko 2017 (downward revision of geological source of methane), and Thornton 2017 (downward revision of ESAS methane emissions by a factor of about 8), ... Please limit self-citation and quote more of the recent literature. I strongly suggest also to include in section 8.1 a short discussion about these recent papers and the implication for your work : you downward estimate of 37 Tg/yr is smaller than the previous 50 Tg/yr, but still well above the Petrenko suggested value of 15 Tg/yr.

3/ The methodology section needs attention. It has to explain more in detail how the actual flux measurements or the statistical approaches are used to build the flux estimation or to point more precisely to sections into the supplementary.

4/ You have to explain more clearly at the beginning that some part may be missing in the gridded map and that it means a possible underestimation of global emissions. I am not convinced by the extrapolation made by the authors to complement the gridded estimate as it mostly rely on very rough estimates of the missing part (some additional areas emitting might be there and there, Arbitrary 50% flux, ...). In this sense column 3 of table 3 is a bit strange to me as roughly estimated whereas you spend a lot of time and energy to properly provide gridded estimates of column 2. This extrapolation has to be presented much more carefully and not put at the same level than the gridded estimate. Also the sentence about some of the emission not being an update/improvement, mentioned several in the paper, is a bit strange to me and should be rephrased. In fine, I would just indicate in the conclusion that the gridded product

will/may be revised regularly, upward or downward, when more data become available

5/ An uncertainty estimate has to be given for emissions of all categories (and reported in table 5), as for MS and isotopic signatures. This is critical for consistency of the paper and usage in atmospheric inversion. Although it might not be easy, the authors are the best choice we have to make such estimates, which else will be made by inverse modellers who probably know much less on the specific topic.

6/ All along the text & tables : please harmonize the number of significant digits in the numbers provided. Considering the uncertainties I am not sure that 3.87 Tg/yr is relevant for instance for OS and I suggest to at least use 3.9 Tg/yr or possibly 4 Tg/yr. No more than 1 digit after the comma in any case.

The paper will definitely be a major (evolving) piece to improve the global methane budget and should be improved after these general comments and the specific ones below are addressed.

Specific comments :

Abstract : "representativeness for many sources" suggested : and their isotopic signatures

Abstract : "This gap is particularly wide for geological CH4 seepage, i.e., the natural degassing of hydrocarbons from the Earth's crust. While geological seepage is widely considered the second most important natural CH4 source after wetlands, it has been mostly neglected in top-down CH4 budget studies, partly given the lack of detailed a priori gridded emission maps". This sentence is polemical and should be removed from the abstract which should reflect the work done. Considering the estimates of the CH4 emissions from geological seepage in the literature and in this paper, and the uncertain estimates from inland water systems, it is difficulet to say robustly that geological source is the 2nd. I would say a major source. And the lack of interest is true for past budgets but recent ones (e.g. Saunois et al., 2016) account for this source.

P2 l5 : I suggest to update the ref to Saunois et al., 2016 and 558 MtCH4/yr

P2 l7 : "emission inventories" and process-based models

P2 l8-9 : TD and BU show strong disagreement only or natural sources, please precise.

P2 l 9-12. The sentence has several problems. Schwietzke et al 2016 is not 3D inverse modelling but box modelling. The improvement brought by recent 3D modelling is arguable the recent study mentioned actually enlarge the range of emission estimates and needs to be further reproduced to pretend to get closer to the truth than other studies. I would rephrase to point that the usages of updated inventories of isotopic signatures has brought new constraints for the global methane budget. In any case, please rephrase.

P2 l28 : geological degassing is today recognised as the second most important natural CH4 source after wetlands : see remark from the abstract. Also, the recent Petrenko paper should be quoted here (and commented later in the paper) as it proposes a downsizing of geological emissions to 15 Mt/yr at maximum.

P2 l 29-30 : it is a bit unfair to quote specific papers when the highly visible synthesis from IPCC or GCP mention geological emissions in thei budget (e.g. Saunois et al., 2016). Please rephrase.

P3 l34 – figure 1 : References to other sources should be updated to the Saunois et al budget (GCP 2nd budget) instead of Kirschke et al. (GCP 1st budget). Please precise that figure 1 reflects literature and not the results of this paper.

P5 l12-13 : what does it mean ? How can you know there is a seep of you cannot locate it ? Please precise and rephrase.

P5 l 22 : Why not documented ? please provide a reason.

Section 4.1 : How can you be sure that oil&gas seeps are not double counted in anthropogenic inventories as possibly located close to fossil fuel exploitation facilities ? It

is important to mention this somewhere in the paper and possibly discuss it as double counting is one clue to explain why bottom-up and top-down studies are not consistent for natural methane emissions.

P5 l 30 : "few tens" : can't you be more precise ? it is important to have a more precise idea of the fraction compared to the total number.

P5 l30-39 : the methodology should be a bit more detailed here (the supplementary does not bring much more on this). How did you use the direct measurements to calibrate ? How did you attribute a measurement to a type of seeps ? how many types did you use ?

P5 l 38 : how do you account for miniseepages ? please provide ref or explanation.

Section 4.2.3 big emitters. What fraction of these big emitters has been directly observed ? It would be important to mention as they are not so numerous and a strategy to refine the estimate would be to measure them all (if not done yet). Please precise here.

Section 4.5.1 : This section needs attention l15-16, if you do not do this work to update or improve estimates, why doing it so ? I am pushing a bit what you write but please rephrase. L28-29 : where does the 30% and 50% come from ? The 50% looks like a bit arbitrary ? is this 100% error reflected in column 3 of table3, moving from 3.8 to 8.1 Tg/yr for OS ? It is not clear to me why producing a gridded map if it cannot be used directly for global scale and needs re-assessment of emissions. Please clarify this section and the meaning of column 3 of table 3.

Section 4.6 : It is strange to me that you do not provide an uncertainty attached to emissions and signature in this section as in 6.6 for MS. "Order of magnitude" means a factor of 10 uncertainty. Does it means that OS emissions range from 0 to 38 Tg/yr ? Please be more precise in this section of possible or explain whay you cannot provide a range or a sigma for uncertainties.

P9 l 27: there is no section 5.5.1.

Section 5.5 : The total of 20Tg/yr has been highly controversial in the past years and recent papers related to ESAS largely reduced emission estimates (Berchet 2016, Thornton 2017). I would not present this number as a target to reach in the text. Lines 15 to 20 are highly arbitrary and should be identified as so. Why 5 to 10 Tg/yr ? These extrapolations should be taken with caution to me and mentioned as so. Again do these estimate refer to column 3 of table 3 (5-12 Tg/yr, where text mentions 7-12) ?

Section 8.1: This section has to be enriched to reflect a more complete spectrum of estimates than the ones provided by the co-author of this paper. At least the estimate from the recent Petrenko 2017 paper is important because it lowers to at maximum 15 MT/yr the total global value of geological emissions. Also, the 14C constraint on total 14C free methane from Lassey 2007 could be quoted. These elements should be quoted and discussed briefly in this section.

Section 5.6 : same remark as for OS : can you provide an uncertainty number for emissions (sigma or range) as in 6.6 for MS ? Section 6 : Even more critical than with OS emissions, the possible double counting with anthropogenic emissions should be addressed. How can we be sure that this diffuse source is not part of the oil&gas estimates of inventories ? OS are precisely located so the risk may be smaller than for diffuse MS. But for diffuse sources in the middle of oil&gas fields it seems more tricky. Please at least mention/discuss this in the text as a cause of uncertainty in section 6.6

P15 l29 : what is the impact of the 4 km choice on the emission estimate ?

P15 l 37 : "few cases" : please provide a more precise number if possible.

P16 l 7 Again this sentence is unclear to me. Please rephrase

P16 l25 : "It is known . . ." : any reference to justify this ? Any explanation ? please provide a reference or explanation

Section 7.6 : as for other categories please provide a number (sigma/range) for the

uncertainty on GM emissions as in 6.6

P17 l34 : again please clarify this sentence.

Table 5 : As already mentioned, please provide an uncertainty estimate for emissions from OS, SS and GM and fill it in table 5, column 3.

P18 l27 : is there a risk that some SH emissions are forgotten because of les knowledge of the terrain ?

P19 l4-6 : The 20 Tg/y value previously widely used for SS has been revised downward by several studies at least because of ESAS region (Berchet 2016, Thornton 2017). As already noticed, one should stop giving the idea that this value is kind of a target to reach, as suggested here and in the corresponding paragraph of the text (see previous comment). The reference given here (Kvenvolden et al. 2001) seems a bit old regarding the past years activity on these emissions. Can the author provide a more recent reference and rephrase according to this remark ?

P19 l31-33 : if no description of geological is given in an inverse modelling exercise, all the flux is spread on other distribution, possibly for onshore emissions, but with no guaranty, on the anthropogenic fossil emissions. So the term low bias should be rephrased (while the high bias is possibly correct for anthropogenic fossil). Please rephrase.

---

## Author Comment (AC2) · 16 Nov 2018

Author comments to Anonymous Referee #2

R: Reviewer A: Author

GENERAL COMMENTS R: According to the abstract the global mean isotopic signature of geologic emissions is -48.5 to -49.4 per mil. In addition to the global average, a global map of fractionation factors is provided. The question is how those numbers were derived.

A: This is explained below (last comment in this page).

R: The section on on-shore seeps mentions (section 4.3) that it is a combination of

[Figure]

measurements and estimates. The measurements are 'as indicated in the literature' without references.

A: It was implicit that the "literature" reporting isotopic data is in the onshore seeps inventory, which is described in section 4.1. This is now clarified in the ms.

R: The estimates follow 3 rules that are listed without specifying which rule applies when.

A: The 3 rules are applied depending on the availability of the data. For clarity, we have rephrased as follows: . . ..or (b) estimated on the basis of isotopic values following one of the following three procedures, in priority order: - of similar seeps occurring in the same basin (when these data are available) - of reservoir gas in the same petroleum field, from Sherwood et al. (2017) dataset or literature - suggested by local petroleum geology (existence of microbial gas, thermogenic gas, oil), when the previous procedures cannot be applied.

R: A global mean is used in regions for which no data exist. It is unclear, however, where this applies.

A: Since onshore seeps are 2827, it is not possible to list in the text the seeps that have no $\delta$13C value, also because the OS inventory cannot be provided due to license restrictions, as indicated in the Data Availability chapter. However, the regions with emission-weighted mean are shown in Figure 10.

R: The descriptions of the other subsections on isotopic information are similarly general, and unspecific.

A: For SS, the isotopic information is given in Table S4, as indicated in the text. For MS, we have described in detail the procedure (section 6.3), and some data are linked to OS inventory (which cannot be released for license restrictions). For GM, the isotopic data are shown in the GM dataset (available at https://doi.org/10.25925/4j3f-he27) and the derivation of the mean values is described in detail (section 7.3).

R: It is unclear how the range between -48.5 and -49.4 should be interpreted. Is this supposed to be an uncertainty range?

A. As indicated in Section 8.2, the value -48.5 is emission-weighted using gridded emissions, the value -49.4 uses global emission estimates. Since the gridded emissions are not complete (as explained in section 8.1) the second value is likely more realistic, as now clarified in the Conclusions (point (e)).

R: Some of the paragraphs dealing with uncertainty in d13C mention 15 per mil as a range of reported numbers. Then how is it possible to arrive at an uncertainty on the global fractionation within 1 per mil?

A: The global uncertainties of Table 5 refer to the uncertainty of global emission-weighted values, not to individual data. This is now clarified with a note at the bottom of the table. The 15 per mil uncertainty was mentioned (in Section 4.6) only for OS, referring to maximum uncertainty of individual estimated data. In Table 5, for OS we reported 1 per mil referring to an approximate expression of the uncertainty of the global emission-weighted mean, which is dominated by 76 big emitters, whose specific uncertainty is 0.1 per mil (see last line of Section 4.6). We adopted then 1 per mil (Table 5) as average order of magnitude of the uncertainty. This is now clarified in Section 4.6.

R: Important for the application of the geologic emission map to inverse modelling is the information on uncertainty that is provided. However, the method that is used to quantify uncertainties is questionable. For example, for some sources (e.g. OS and SS) it is stated that the uncertainty in the geographical distribution is practically zero. While this may apply to specific sources that have been located, there should be uncertainty from sources that have not yet been found. The question then becomes to what fraction of the emissions this may apply.

A: Uncertainties are now better described and assessed for all geo-sources (see additions in sections 4.6, 5.6 and 7.6). Of course, we refer only to the uncertainty of the

data provided. If we do not know whether and where other sources exist, how can we define an uncertainty?.

R: Section 4.6 on onshore seeps describes contributions to uncertainty, but does not provide a single number. For the uncertainty in the isotopic fractionation factor a maximum difference between estimates is mentioned, but it is unclear how representative that maximum is. Table 5 mentions ±1 per mil, which seems quite accurate, but it is unclear where this number comes from.

A: OS uncertainties are now better described and assessed (see additions in section 4.6).

R: In the description of GM uncertainties, the emission estimates are discussed without providing a clue of what the uncertainties may be. To summarize: The treatment of uncertainties requires several clarifications.

A: GM uncertainties are now better described and assessed (see additions in section 7.6).

R: If I understand correctly the gridded emission maps do not account for the global extrapolation, i.e. they account for 37 Tg/yr of methane emissions.

A: right.

R: For the remainder no specific information is available, which raises the question what the extrapolation of almost a quarter of the emissions is based on. The assumptions underlying the extrapolation are not spelled out, which is critical not only for trying to understand how they were derived, but also to guide users of the emission dataset as to where missing emissions are to be expected (see my earlier remark about the uncertainty in the spatial distribution of the emissions). Besides a clearer description of the assumptions underlying the extrapolation, some discussion is needed of what are reasonable assumptions (in terms of geographical regions) to put the remaining sources or to account for their uncertainty.

A: The extrapolated emissions (and related assumptions) are explained in Table 3 note. Extrapolated emissions include factors that cannot be accounted in the gridding, such as MV eruptions, the OS foreseen but not included in the gridded inventory, unidentified or not-investigated offshore seepage. Geographical regions of these extra emissions (difference between gridded and extrapolated emission) can be understood reading the specific Sections describing the distribution of OS, SS, MS and GM.

R: Some discussion is needed of how to distribute the emissions not only spatially but also temporally. Without this information, modelers will probably assume that all emissions are constant over the year. Besides eruptions, for which you would obviously need to know the timing, continuous emissions may vary with environmental conditions (temperature, soil water content?). A few sentences of discussion would be useful to provide information to guide the choice of temporal distribution, and the uncertainty assigned to it (for atmospheric modelers it is relevant to know within what bounds continuous emission may vary over the year)

A: This is a good point. We were actually planning to add a short section discussing temporal geo-emission variations, also with reference to recent works reporting estimates of geo-CH4 emissions in the geological past (Younger Dryas and Preboreal intervals). The new section is added in the final revised version of the manuscript.

SPECIFIC COMMENTS R: Page 3, line 10: What are 'originally ad hoc developed datasets'? Which parts of the inventory are based on such data?

A: Everything is described in Section 4, 5, 6 and 7. This is now clarified in Introduction.

R: Page 4, line 8: '. . . reported in the supplement'. Add '(S6)'. OK

Page 4, line 14: How were the 'single OS, SS, MS, and GS shapefiles' derived? Aren't these just lists of coordinates of reported seeps? If so, this needs to be made clear to avoid the impression that unspecified information went into these shape files.

A: The shapefiles were in point or polygon format depending on the type of the source.

We have now added the following text for clarity:

Geo-CH4 emission and isotope datasets were imported in ArcGIS environment and saved in either point (OS and GM) or polygon (SS and MS) shapefile format, including coordinates and attributes (i.e., type of emission, area, emission factor, isotopic CH4 values, plus geographical information, such as country and region).

R: Page 5, line 13: '. . . listed without coordinates' Does this mean that no geographical information is provided at all?

A: Only the country or rough indication of the basin (region) is reported.

R: Page 5, line 15: 'The total number of 3439 OS represents about 30% . . .' Does this mean that the remainder is part of the 'global extrapolation'? This question applies as well to the 50% mentioned later in this paragraph.

A: yes, and this is indicated in Table 3.

R: Page 5, line 33: 'theoretical values were used'. The explanation that follows makes clear what these values account for, but it is unclear what the values are and based on which criteria they are assigned.

A: The values are reported in Table S1, as indicated in the text. The value assignment is based on experimental data (measured fluxes) and flux modelling (mainly depending on seep size) reported in various papers, listed in Etiope (2015). This is now clarified in the text.

Page 5, line 38: What is the difference between 'micro' and 'mini' seepage?

Microseepage is the diffuse seepage, independent of macro-seeps. Miniseepage is the diffuse seepage surrounding a macro-seep. Definitions are given in Etiope (2015) and this is now indicated in the text. In Section 2 we outlined that, for details on seepage processes and terminology, the reader may refer to a series of fundamental works.

R: Page 6, line 38: As discussed on Etiope et al, 2009, the isotopic signature of the

reservoir may be a poor indicator of the isotopic signature of the emissions due to fractionation due to advective segregation. In light of this, what is justifying the use of the reservoir signature here?

A: The reviewer has misinterpreted Etiope et al 2009. Advective segregation produces molecular fractionation but not isotopic fractionation. Etiope et al 2009 shows, in fact, that generally there is no substantial difference between reservoir and seep $\delta$13C values.

R: Page7, line 8: '. . . because of multiple counting of 57 seeps . . .' but I thought the point of using ArcGIS was to deal with this kind of issues. Why not assign the emission proportional to area or something like that? There must have been an easy way to avoid double counting.

A: To our knowledge there is no way to assign the value of the point, located on the boundary of two or four cells, to one specific cell. The only way would be to shift the point, changing its coordinates. We preferred to avoid this procedure and keep the original coordinates, as the double counting is actually negligible compared to the uncertainty of the total emission.

R: Page 7, line 16: '. . . OS grids are not meant to update or refine . . .' but the reference is to a paper that was published 10 years ago. Table 1 lists data sources for category OS that are of more recent dates. What does this statement mean for those updates?

A: Those updates basically refer to the inventory (existence and location) of seeps, not necessarily to flux values. Almost all OS flux values for the grid are theoretically derived (for the reasons described in Section 4.2.1), so OS gridding cannot be considered an advance of previous OS emission estimates procedures.

R: Page 7, line 22: '. . .those estimates are indicated in the Table as upper limits . . .' Are marine MV larger emitters as onshore MV's? Otherwise I don't understand

this statement. It seems not obvious to me, since a fraction of the emissions from submarine MV's will be oxidized in the water column.

A: Shallow marine MV can be important emitters, but regardless, formally, for the comparison with gridded emissions, they should be considered as part of SS. The problem is that Dimitrov, 2003 and Milkov et al. 2003 did not distinguish clearly the emission from submarine MV and that from onshore MV. This is now clarified in section 4.5.1.

R: Page 8, line 10: The total mean value is just the average of all reported d13C values? A: YES

R: Since emission weighting seems such an obvious improvement of this estimate why are both estimates mentioned?

A: This is done to show how big emitters control the weighted-emission mean, leading to a final value of -46.6 per mil.

R: Page 8, line 35: '< 500 m deep . . . McGinnis et al, 2006' According to McGinnis it is unlikely that seeps from deeper than 100m can contribute significant amounts of methane to the atmosphere. Therefore, the threshold should be 100m instead of 500m.

A: We actually used a misleading reference. McGinnis et a. (2006) is just one of the several studies reporting variable depths on submarine seeps reaching the atmosphere. The 100 m threshold indicated by McGinnis et al (2006) is a minimum value, valid for some observations in the Black Sea, and it neglects bubble plume induced upwelling and excludes plume dynamics of large gas releases, as outlined in successive modeling (e.g., Schmale et al. 2009, doi: 10.1016/j.jmarsys.2009.10.003; Yamamoto et al 2009, doi:10.1016/j.epsl.2009.05.026) and direct observations (Solomon et al. 2009; doi: 10.1038/NGEO574). Greinert et al (2006, doi:10.1016/j.epsl.2006.02.011) and Solomon et al (2009) reported methane bubble columns ascending for >1000 m and >500m respectively. We used therefore an approximate maximum threshold of 500 m (<500 m). We now refer to Solomon et al (2009) in the revised manuscript.

R: Page 11, line 34: 'The similar order of magnitude . . . log normal behavior' If the mean and median are the same than this suggests rather a normal distribution. I don't see why the mean and median in the same order would point to a log normal distribution.

A: We wrote "geometric mean" not "mean".

R: Page 13, line 23: Implicit here is that the shallower reservoirs have a heavier isotopic signature than a deep reservoir. Is this generally true? If so, then why?

A: NO, the opposite. Shallower reservoirs have a lighter isotopic signature. For better understanding, the sentence has been rephrased as follows: ....the MS gas may actually come from shallower reservoirs, not necessarily or not dominantly from the deep productive reservoirs, which are more frequently the literature source of the isotopic value. Therefore, in some cells the real isotopic value could be lighter than that used in the grid maps.

R: Page 14, line 2: What explains the _10% difference between gridded and reported area in this case?

A: The difference depends on the cell size and it increases when cell size increases. The perfect match would be only when the grid cells are infinitely small.

Page 14, line 37: 'buffer applied to individual seeps' I don't understand what this means. A: This is explained in Supplement S3.2 (as now indicated in the text).

R:The reference should probably be to section 6.2.1. A: No, S3.2 (now corrected).

R: Page 15, line 16: The result of 2 different ways of averaging does not sound as a reliable estimator of uncertainty. This seems confirmed by 2 per mil being very small given the range of the fractionation values.

A: We considered it as "approximate expression of the uncertainty"; it is not easy to assess more precisely the uncertainty on those type of data. See also responses to

Reviewer #1 above.

R: Page 16, line 6: 'Accordingly, the total emission estimate . . . (Etiope et al, 2008)' Why is this? Because the accounting approach that is adopted here is not considered meaningful?

A: Simply because accounting approach adopted is not better than previous estimates. Our gridding operation had a specific objective, which is not necessarily the one of improving global emission estimates.

R: Page 17, line 4: What explains the exceptionally heavy isotopic signature of GM emissions?

A: Geothermal methane includes both over-mature thermogenic and abiotic gas, and both are typically 13C-enriched.

R: Page 18, line 2-5: If I understand correctly, the emission maps are referred to here as gridded emissions. It means that they only account for 37 Tg/yr.

A: YES

R: Page 18, line 10: The combined d13C uncertainty depends on the individual uncertainties. The more important question of how they are combined should also be answered.

A: Their combination is explained in the gridding procedure, Section 8.1

Page 19, line 23-24: 'It is expected that using the updated . . . (all else equal)' I don't understand why this would be the case. At -49 per mil atmospheric 13C is really insensitive to geological emissions as it is so close to the mean atmospheric composition.

A: The global source attribution (top-down either via simple box models or via 3D inverse models) includes all methane sources, whether their isotopic source signatures are "close" or "not close" to the atmospheric signal. In fact, these models quantitatively

estimate the magnitude of each source based on how close (on a continuum) each source is from the atmospheric signal.

Page 19, line 32-34: It is unclear to me why this would be the case (see my previous point).

A: Previous top-down studies have lumped fossil fuel industry and natural geologic seepage together for two reasons: (i) Source signatures were assumed to be the same, and (ii) source locations largely overlap. Hence, it has been difficult for any top-down model to distinguish the two in the absence of an a priori natural geologic seepage map. Additionally, global methane emissions from natural geologic seepage have been assumed to be minor in many previous top-down studies. As a result, almost the entire sum of both sources was previously attributed to fossil fuel industry.

R: Suppl. Page 9: '. . . considered for the text file' Which text file?

A: the csv MS grid file (now clarified).

R: Suppl. Fig S3: To what extend could the difference in slope between the two regression lines be explained by the use of the erroneous syringe method for the larger MV's (increasing the micro seepage would bring the lines closer together)

A: Difficult to say because both small and large MV data are underestimated

R: Suppl. Page 8: 'tested' or 'evaluated' i.o. 'checked'. The latter suggests that the validity of the sensitivity was verified using some external information, which is not the case.

A: OK, tested

TECHNICAL CORRECTIONS Page 6, line 31: 'emission' i.o. 'output' Page 8, line 37: 'emission' i.o. 'output' Page 6, line 24: 'Grubbs' i.o. 'Grubs' Table S3, caption: 'Azerbaijan' i.o. 'Azerbaiajn'

OK, all technical corrections have been done.

We thank the anonymous reviewer for the valuable and careful revision of the work

---

## Author Comment (AC3) · 16 Nov 2018

Author reply to Reviewer #3

R: Reviewer A: Author

R: 2/ You quote 20 self publications (Etiope or Etiope et al). It seems a bit too much regarding the total number of references and I recommend to keep only the main ones.

A: We really tried to use self-citations as little as possible, but the cited papers cannot be skipped as they refer to key reviews or are source of data used in the present work. Only two references could be deleted. References for specific seeps are reported in the inventories described in section 11.

[Figure]

R: Also, some recent relevant references are missing such as Petrenko 2017 (downward revision of geological source of methane), and Thornton 2017 (downward revision of ESAS methane emissions by a factor of about 8). I strongly suggest also to include in section 8.1 a short discussion about these recent papers and the implication for your work : you downward estimate of 37 Tg/yr is smaller than the previous 50 Tg/yr, but still well above the Petrenko suggested value of 15 Tg/yr.

A: We actually planned to add a discussion on Petrenko et al 2017 estimates in the revised version, as part of a chapter dealing with temporal variations of geological sources. This is now the new Section 9. We have added Thornton et al 2016; we used Berchet et al 2016 as estimate of ESAS emission. It is not dissimilar from the estimate suggested by Thornton et al, but it seems to have considered a wider ESAS area than the one measured by Thornton et al.

R: 4/ You have to explain more clearly at the beginning that some part may be missing in the gridded map and that it means a possible underestimation of global emissions.

A: ok a sentence is now added at the end of Introduction

R: I am not convinced by the extrapolation made by the authors to complement the gridded estimate as it mostly rely on very rough estimates of the missing part (some additional areas emitting might be there and there, Arbitrary 50% flux, : : :).

A: The extrapolation only refers to the global estimate (for completeness of the paper and the obvious need to compare the gridded emissions with published global total estimates), and the limitations of the approach are discussed in the paper. The extrapolation has no impact on the gridded emission. For inverse modelling, the gridded emissions will be used.

R: In this sense column 3 of table 3 is a bit strange to me as roughly estimated whereas you spend a lot of time and energy to properly provide gridded estimates of column 2. This extrapolation has to be presented much more carefully and not put at the same

level than the gridded estimate.

A: The extrapolation has the only scope to show that gridded emissions do not necessarily represent the actual global emission, because the datasets developed for the gridding may not be complete or may not contain the information necessary for improving the estimates (as happened for oil-gas seeps, SS and GM). This is now clarified at the end of Introduction and in Section 8.1.

R: In fine, I would just indicate in the conclusion that the gridded product will/may be revised regularly, upward or downward, when more data become available.

A. We agree, and this is stated at the end of Conclusions.

R: 5/ An uncertainty estimate has to be given for emissions of all categories (and reported in table 5), as for MS and isotopic signatures. This is critical for consistency of the paper and usage in atmospheric inversion. Although it might not be easy, the authors are the best choice we have to make such estimates, which else will be made by inverse modellers who probably know much less on the specific topic.

A: Yes, due to the nature of the data and derivation of the emission factors, uncertainty in final numbers is not of direct derivation and requires assumptions or arbitrary evaluations; this is why we preferred to report the factors controlling the uncertainty. However, we have now extended the discussion and provided approximate uncertainty values.

R: 6/ All along the text & tables : please harmonize the number of significant digits in the numbers provided. Considering the uncertainties I am not sure that 3.87 Tg/yr is relevant for instance for OS and I suggest to at least use 3.9 Tg/yr or possibly 4 Tg/yr. No more than 1 digit after the comma in any case.

A: Sure, one digit must be used.

Specific comments : R: Abstract : "representativeness for many sources" suggested : and their isotopic signatures

A: OK

R: Abstract : "This gap is particularly wide for geological CH4 seepage, i.e., the natural degassing of hydrocarbons from the Earth's crust. While geological seepage is widely considered the second most important natural CH4 source after wetlands, it has been mostly neglected in top-down CH4 budget studies, partly given the lack of detailed a priori gridded emission maps". This sentence is polemical and should be removed from the abstract which should reflect the work done.

A: These sentences explain the motivation for this work, and are therefore central to this study. They are not meant to devalue previous top-down studies, and we tried to reflect this in the language provided.

R: Considering the estimates of the CH4 emissions from geological seepage in the literature and in this paper, and the uncertain estimates from inland water systems, it is difficulet to say robustly that geological source is the 2nd. I would say a major source.

A: Geological emission is formally reported as 2nd CH4 source by the latest IPCC report (Ciais et al 2013). However, ok for using a more moderate "major source" expression.

R: And the lack of interest is true for past budgets but recent ones (e.g. Saunois et al., 2016) account for this source.

A: We do not mention a "lack of interest" but just the impossibility to use properly this source in top-down procedures because of the lack of priori gridded emission maps. We have rephrased as follows:

While geological seepage is widely considered a major source of atmospheric CH4, it has been largely neglected in 3D inverse CH4 budget studies given the lack of detailed a priori gridded emission maps.

R: P2 l5 : I suggest to update the ref to Saunois et al., 2016 and 558 MtCH4/yr A: OK

[Figure]

R: P2 l7 : "emission inventories" and process-based models A: OK

R: P2 l8-9 : TD and BU show strong disagreement only or natural sources, please precise. A: OK

R: P2 l 9-12. The sentence has several problems. Schwietzke et al 2016 is not 3D inverse modelling but box modelling. The improvement brought by recent 3D modelling is arguable the recent study mentioned actually enlarge the range of emission estimates and needs to be further reproduced to pretend to get closer to the truth than other studies. I would rephrase to point that the usages of updated inventories of isotopic signatures has brought new constraints for the global methane budget. In any case, please rephrase.

A: Rephrased as follows: "Global box-modelling based on isotopic measurements (stable C isotope ratio, $\delta$13C-CH4) of source signatures and the atmosphere combined with three-dimensional (3D) forward modelling using trends and spatial gradients recently improved the knowledge of major sources (fossil-fuel, agriculture and wetlands) and their spatio-temporal variation (e.g., Schwietzke et al 2016).

R: P2 l28 : geological degassing is today recognised as the second most important natural CH4 source after wetlands : see remark from the abstract. Also, the recent Petrenko paper should be quoted here (and commented later in the paper) as it proposes a downsizing of geological emissions to 15 Mt/yr at maximum.

A: OK for major source, but Petrenko et al refers to a late Pleistocene emission estimate, which can be used as reference for today's emission only assuming that geoemissions are constant over time (which is not true). However, Petrenko et al is now discussed in a specific Section 9.

R: P2 l 29-30 : it is a bit unfair to quote specific papers when the highly visible synthesis from IPCC or GCP mention geological emissions in thei budget (e.g. Saunois et al., 2016). Please rephrase.

A: Ok, Saunois et al 2016 is added.

R: P3 l34 – figure 1 : References to other sources should be updated to the Saunois et al budget (GCP 2nd budget) instead of Kirschke et al. (GCP 1st budget). Please precise that figure 1 reflects literature and not the results of this paper.

A: OK, done

R: P5 l12-13 : what does it mean ? How can you know there is a seep of you cannot locate it ? Please precise and rephrase.

A: Because their existence, as number of seeps, is reported but their location is not provided.

R: P5 l 22 : Why not documented ? please provide a reason.

A: we suppose that many seeps in Africa and S.America are not documented because of the paucity of works (addressed to seeps) in these regions. It is clarified now.

R:Section 4.1 : How can you be sure that oil&gas seeps are not double counted in anthropogenic inventories as possibly located close to fossil fuel exploitation facilities? It is important to mention this somewhere in the paper and possibly discuss it as double counting is one clue to explain why bottom-up and top-down studies are not consistent for natural methane emissions.

A: But anthropogenic (fugitive emission) emissions are estimated by process-based modelling and specific emission factors. We do not see how there may be double counting with a natural phenomenon driven by different processes and with different emission factors. Double counting may happen with remote sensing (air, satellite) or micrometeorological (eddy-covariance) techniques, but these are not used for global emission estimates of fossil fuel. In fact, the gridded maps of this paper may help avoid double-counting in top-down studies by providing for the first time a geological seep grid that can be overlaid with a fossil fuel industry grid. Atmospheric measurements can then better constrain the sum of both while both grids help attribute sources.

R: P5 l 30 : "few tens" : can't you be more precise ? it is important to have a more precise idea of the fraction compared to the total number.

A: We estimated a total of about 100-200 seeps; it is not possible to be more precise as some papers (e.g., Walter Anthony et al. 2012) do not specify the number of seeps where the flux was measured. We have however referred to a summary table reported in Etiope (2015).

R: P5 l30-39 : the methodology should be a bit more detailed here (the supplementary does not bring much more on this). How did you use the direct measurements to calibrate ? How did you attribute a measurement to a type of seeps ? how many types did you use ?

A: The value assignment is based on experimental data (measured fluxes) and flux modelling (mainly depending on seep size) reported in various papers, listed in Etiope (2015). This is now clarified in the text. We do not think it is appropriate to include in this paper all technical (gas-geochemical) details (how a seep flux is calibrated, etc.), as it is outside the scope of the work.

R: P5 l 38 : how do you account for miniseepages ? please provide ref or explanation.

A: A reference was given.

R: Section 4.2.3 big emitters. What fraction of these big emitters has been directly obvserved? It would be important to mention as they are not so numerous and a strategy to refine the estimate would be to measure them all (if not done yet). Please precise here.

A: Emission from the big emitters (almost all mud volcanoes) is estimated using the emission factor and area approach described in Section 4.2.2. This is now indicated.

R: Section 4.5.1 : l15-16, if you do not do this work to update or improve estimates, why doing it so ? I am pushing a bit what you write but please rephrase.

A: Gridding has the scope to provide a priori maps for inverse modelling, as indicated in the Introduction. The datasets developed for the gridding may not contain the information necessary for improving the estimates. See also our response above: The extrapolation only refers to the global estimate (for completeness of the paper and the obvious need to compare the gridded emissions with published global total estimates), and the limitations of the approach are discussed in the paper. The extrapolation has no impact on the gridded emission. For inverse modelling, the gridded emissions will be used.

R: L28-29 : where does the 30% and 50% come from ? The 50% looks like a bit arbitrary ?

A: As stated in the text, 30% comes from knowing the total number of seeps on Earth (as discussed in Section 4.1). Yes, 50% is arbitrary, and we use hypothetical terms "may contribute at least 50% of the global emission..", "...could...".

R: is this 100% error reflected in column 3 of table3, moving from 3.8 to 8.1 Tg/yr for OS ?

A: Only partially. About 3.1 Tg/y are due to mud volcano eruptions, not accountable in the gridding. All is explained in Table 3 footnote.

R: It is not clear to me why producing a gridded map if it cannot be used directly for global scale and needs re-assessment of emissions. Please clarify this section and the meaning of column 3 of table 3.

A: The grid can be used for global modeling, but the represented global flux therein may be incomplete. We have explained above that the datasets developed for the gridding may not contain be complete and/or may not have the information necessary for improving the estimates. Gridding has the scope of providing a priori maps for inverse modelling. Note that the same may be true for other published and widely used natural and anthropogenic CH4 flux grids. For example, different wetland flux

grid products vary widely in their spatial distribution due to different data sources used (which may not be complete either). The geological seep maps developed here are unique in the sense that they are the first comprehensive product of this source, and thus no ensemble inverse runs are possible (like it is for wetlands). Anthropogenic grids are similar in the sense that it is very unlikely that every landfill in the world is included.

R: Section 4.6 : It is strange to me that you do not provide an uncertainty attached to emissions and signature in this section as in 6.6 for MS. "Order of magnitude" means a factor of 10 uncertainty. Does it means that OS emissions range from 0 to 38 Tg/yr ?

A: This section refers to emissions (and uncertainties) of individual seeps, not of their global emission (which is now determined and discussed).

R: Please be more precise in this section of possible or explain whay you cannot provide a range or a sigma for uncertainties.

A: This is now better explained in the text.

R: P9 l 27: there is no section 5.5.1. A: Corrected

R: Section 5.5 : The total of 20Tg/yr has been highly controversial in the past years and recent papers related to ESAS largely reduced emission estimates (Berchet 2016, Thornton 2017). I would not present this number as a target to reach in the text.

A: We agree and we do not mean to use 20 Tg/y (Kvenvolden et al) as a target, but only as reference. This is now clarified in the text. Anyway, there are no updated global emission estimates for submarine seepage after that value. Berchet and Thornton papers refer only to ESAS and their finding do not imply that Kvenvolden et al estimate was wrong.

R: Lines 15 to 20 are highly arbitrary and should be identified as so. Why 5 to 10 Tg/yr ? These extrapolations should be taken with caution to me and mentioned as so.

A: Of course, these are just hypothetical, potential numbers. We rephrased however

as follows: "...it is plausible that global SS emission exceeds 5 Tg yr-1".

R: Again do these estimate refer to column 3 of table 3 (5-12 Tg/yr, where text mentions 7-12) ?

A: Yes, but now rather than a range, a lower value is indicated (>5 Tg/y, i.e., >5 + 2 including Berchet's upper limit)

R: Section 8.1: This section has to be enriched to reflect a more complete spectrum of estimates than the ones provided by the co-author of this paper. At least the estimate from the recent Petrenko 2017 paper is important because it lowers to at maximum 15 MT/yr the total global value of geological emissions. Also, the 14C constraint on total 14C free methane from Lassey 2007 could be quoted. These elements should be quoted and discussed briefly in this section.

A: We actually planned to add a discussion on Petrenko et al 2017 estimates in the revised version, as part of a chapter dealing with temporal variations of geological sources (now Section 9). As noted above, however, Petrenko et al refers to a late Pleistocene emission estimate, which can be used as reference for today's emission only assuming that geo-emissions are constant over time (which is not true).

R: Section 5.6 : same remark as for OS : can you provide an uncertainty number for emissions (sigma or range) as in 6.6 for MS ?

A: The uncertainty of Berchet's ESAS (2 Tg/y as average) is 2 Tg/y. The uncertainty of other SS (1 Tg/y), from literature, cannot be assessed, because that literature does not provide uncertainties (as clearly indicated in the text). We may use arbitrarily 10% of uncertainty for the 1 Tg/y, so overall uncertainty would be 2.1 Tg/y. If a number is wanted, this can be given.

R: Section 6 : Even more critical than with OS emissions, the possible double counting with anthropogenic emissions should be addressed. How can we be sure that this diffuse source is not part of the oil&gas estimates of inventories ? OS are precisely

located so the risk may be smaller than for diffuse MS. But for diffuse sources in the middle of oil&gas fields it seems more tricky. Please at least mention/discuss this in the text as a cause of uncertainty in section 6.6

A: Similar to our previous comment concerning double counting with OS: anthropogenic (fugitive emission) emissions are estimated by process-based modelling and specific emission factors. We do not see how there may be double counting with a natural phenomenon. Double counting may happen with remote sensing (air, satellite) or micrometeorological (eddy-covariance) techniques, but these are not used for global emission estimates of fossil fuel.

R: P15 l29 : what is the impact of the 4 km choice on the emission estimate ?

A: The 4 km choice does not influence the overall emission estimate, it is only a parameter guiding the gridding, as it served to convert the point data into more realistic areal data (polygons). This is now clarified in the text (section 7.1).

R: P15 l 37 : "few cases" : please provide a more precise number if possible.

A: OK, (<100 sites) is now indicated

R: P16 l 7 Again this sentence is unclear to me. Please rephrase

A: See our comments above. We rephrased as follows: " Although the GM emission grid developed here is expected to improve global CH4 inverse modeling (as it includes previously neglected GM sources), the total GM emission estimate suggested by the gridding, because of the uncertainty of the theoretical emissions, is not meant to update or refine the previous global GM emission estimate (derived by process-based modelling; Etiope, 2015).

R: P16 l25 : "It is known : : :" : any reference to justify this ? Any explanation ? please provide a reference or explanation

A: Ok reference + brief explanation added.

R: Section 7.6 : as for other categories please provide a number (sigma/range) for the uncertainty on GM emissions as in 6.6

A: This is now provided.

R: P17 l34 : again please clarify this sentence.

A: Same comment above (why gridding could improve emission estimates only for some types of emission)

R: Table 5 : As already mentioned, please provide an uncertainty estimate for emissions from OS, SS and GM and fill it in table 5, column 3. A: Done for all

R: P18 l27 : is there a risk that some SH emissions are forgotten because of les knowledge of the terrain ? A: Yes.

R: P19 l4-6 : The 20 Tg/y value previously widely used for SS has been revised downward by several studies at least because of ESAS region (Berchet 2016, Thornton 2017).

A: See responses above. This is not correct. There is no downward revision of the global SS emission. Berchet and Thornton papers refer only to ESAS and their finding do not imply that Kvenvolden et al global estimate was too high.

R: As already noticed, one should stop giving the idea that this value is kind of a target to reach, as suggested here and in the corresponding paragraph of the text (see previous comment). The reference given here (Kvenvolden et al. 2001) seems a bit old regarding the past years activity on these emissions. Can the author provide a more recent reference and rephrase according to this remark ?

A: No, there is no updated global estimate for SS emission. We have however clarified that Kvenvolden et al number is not a target to reach, but just a theoretical reference for the gridded estimate.

R: P19 l31-33 : if no description of geological is given in an inverse modelling exercise,

all the flux is spread on other distribution, possibly for onshore emissions, but with no guaranty, on the anthropogenic fossil emissions. So the term low bias should be rephrased (while the high bias is possibly correct for anthropogenic fossil). Please rephrase.

A: We re-phrased as follows: In the absence of a comprehensive gridded geological CH4 seepage product, global or regional inverse model studies would erroneously attribute a low-bias to CH4 emissions from geological seepage. This is because of a de-facto zero geological a priori estimate. At the same time, the inverse studies would erroneously attribute a high-bias to CH4 emissions from fossil fuel industry activity (and potentially other sources) while correctly reporting total emissions of all sources.

We thank the anonymous reviewer for the valuable and careful revision of the work